# Separations in the Representational Capabilities of Transformers and Recurrent Architectures

**Satwik Bhattamishra**[1†]   **Michael Hahn**[2]   **Phil Blunsom**[1,3]   **Varun Kanade**[1†]

[1]University of Oxford   [2]Saarland University   [3]Cohere

## Abstract

Transformer architectures have been widely adopted in foundation models. Due to their high inference costs, there is renewed interest in exploring the potential of efficient recurrent architectures (RNNs). In this paper, we analyze the differences in the representational capabilities of Transformers and RNNs across several tasks of practical relevance, including index lookup, nearest neighbor, recognizing bounded Dyck languages, and string equality. For the tasks considered, our results show separations based on the size of the model required for different architectures. For example, we show that a one-layer Transformer of logarithmic width can perform index lookup, whereas an RNN requires a hidden state of linear size. Conversely, while constant-size RNNs can recognize bounded Dyck languages, we show that one-layer Transformers require a linear size for this task. Furthermore, we show that two-layer Transformers of logarithmic size can perform decision tasks such as string equality or disjointness, whereas both one-layer Transformers and recurrent models require linear size for these tasks. We also show that a log-size two-layer Transformer can implement the nearest neighbor algorithm in its forward pass; on the other hand recurrent models require linear size. Our constructions are based on the existence of $N$ nearly orthogonal vectors in $O(\log N)$ dimensional space and our lower bounds are based on reductions from communication complexity problems. We supplement our theoretical results with experiments that highlight the differences in the performance of these architectures on practical-size sequences.

## 1   Introduction

Transformers [59] are the go-to architecture for building LLMs [11], but recently, there has been significant interest in reviving recurrent architectures to build LLMs for practical tasks [21, 42, 45] to circumvent the quadratic complexity of inference for Transformers. While Transformers process all tokens in parallel, maintaining $N$ vectors, and are in some sense stateless, recurrent models primarily store information in a fixed-size hidden state that they can update during the course of the computation. This raises a fundamental question that we explore: Are there tasks that are computationally easier for one architecture to represent but significantly more difficult for the other one?

Ever since Transformers supplanted LSTMs, there has been significant interest in understanding the differences in the computational capabilities of the models both theoretically and empirically. On the one hand, even as Transformers have been widely adopted for building LLMs, several studies found that they struggled at modeling various formal languages, particularly those that required modular counting or state-tracking [5, 16, 38]. At the same time, despite substantial effort, it has proven hard to match the performance of Transformer-based LLMs at scale with recurrent architectures [15, 45, 21]; in particular, it has been observed that LLMs based on recurrent models struggle on associative recall or extraction-related tasks [15, 2]. More recently, Arora et al. [2] and Bhattamishra

---

[†]Corresponding Authors: satwik.bmishra, varun.kanade@cs.ox.ac.uk

38th Conference on Neural Information Processing Systems (NeurIPS 2024).

et al. [8] demonstrated that Transformers are better than attention-free models at synthetic tasks based on associative recall and more general forms such as implementing nearest neighbors. Based on these observations, it is natural to wonder if such tasks are theoretically easier for Transformers to represent in comparison to recurrent models.

**Our Contributions.** We show differences in the representational capabilities of Transformers and recurrent models across several natural tasks of real-world relevance. In this paper, we show strong *separation* results: when a task is easy we show that it can be expressed by one architecture with size poly-logarithmic in the input length $N$; on the other hand we show the other type of architecture requires the size to be linear in $N$. We describe the tasks studied and our key results below.

**(i) Index Lookup.** Given a sequence of symbols $s_1, \ldots, s_N$ followed by a position $p \in [N]$, a model has to output the symbol in the $p$-th position $s_p$ (Figure 1a). Our first result shows that one-layer Transformers with size poly-logarithmic in $N$ can express this task (Theorem 1) whereas any recurrent model must have width $\Omega(N)$ to perform this task (Theorem 3).

**(ii) Bounded Dycks.** Dyck languages with bounded depth require a model to recognize whether a string of parentheses is well-balanced (Figure 1b). These formal languages have received a great deal of attention in the study of neural sequence models because they capture hierarchical dependencies that occur in natural languages [17, 25, 23, 6]. They are also central to formal language theory as all context-free languages can be expressed in terms of Dyck languages [14]. In contrast to the results for index lookup, we show that one-layer Transformers must have a size that grows linearly in input length to represent this task (Theorem 5), whereas prior works found that constant-size recurrent models can express this task [25, 6].

**(iii) String Equality.** This task is formalized by the Equality function $\mathrm{EQ}(\mathbf{x}, \mathbf{y}) = \mathbb{I}[\mathbf{x} = \mathbf{y}]$ where $\mathbf{x}, \mathbf{y}$ are two strings of length $N$; it models the natural task of matching two documents. We show that log-sized two-layer Transformers can represent the function EQ (Theorem 6), whereas both one-layer Transformers (Theorem 12) and recurrent models (Theorem 11) must have a size that grows linearly with the sequence length. These results extend to a broader class of Boolean functions.

**(iv) Nearest Neighbor and Associative Recall.** In this task, a model is provided with a sequence of inputs and labels $\mathbf{x}_1, y_1, \ldots, \mathbf{x}_{k-1}, y_{k-1}$ followed by a query input $\mathbf{x}_k$. The model has to determine the label $y_k$ of the query input by applying the nearest neighbor algorithm in its forward pass (Figure 1c). This task subsumes various associative recall tasks considered in earlier works (cf. Appendix G) and was introduced to understand in-context learning. Bhattamishra et al. [8] found empirical differences in the performance of Transformers and attention-free architectures, though theoretical understanding is lacking. We show that a two-layer Transformer with size logarithmic in the input length can perform this task (Theorem 7), whereas recurrent models require a size linear in the length (Theorem 8).

We also empirically investigated the performance of Transformers and standard recurrent models [26], including recently proposed state-space models [22, 21] on these tasks. The observed behavior is along the lines indicated by our theoretical results.

**Our Techniques**. For constructing Transformers that perform the tasks, one of the key requirements is the ability to precisely attend to specific positions. In order to do this with low-dimensional embeddings, our constructions make use of nearly orthogonal vectors obtained using the Johnson-Lindenstrauss lemma [29]; furthermore these can be generated efficiently in logarithmic space which allows the size of the models to be poly-logarithmic in the input length (cf. Appendix B.3).

For our lower bounds, we appeal to results from communication complexity. We show how to obtain protocols with communication complexity bounded by the size of the models. Together with established (and new) communication complexity lower bounds, we obtain lower bounds on the model size. For our lower bound for one-layer Transformers, we derive a lower bound on the communication complexity for bounded Dyck languages (Lemma 1).

## 1.1 Related Work

**Expressivity of Sequence Models.** The expressive power of neural sequence models has been an active area of research, targeting both RNNs and Transformers [e.g. 23, 52, 40, 55, 37]. With parameters of infinite precision, Transformers are universal function approximators [67] and Turing complete [47, 7]. In bounded precision settings, they relate to Boolean circuits [39, 24] and logical

formalisms [12]. Importantly, such studies usually consider asymptotic expressivity of a single model in the limit of unboundedly long inputs. Our study provides a more fine-grained and realistic picture by accounting for the size of the network, and its scaling with the input. The closest to our work is Sanford et al. [51], who first used communication complexity to prove lower bounds for Transformers and RNNs for abstract tasks such as a sparse averaging task and pair/triple detection tasks. Our work extends that line of work to show separations on natural tasks of practical and real-world relevance.

**Formal languages and Algorithmic tasks.** A strand of work has sought to understand sequence models via empirical analysis on formal languages and algorithmic tasks [5, 57, 68]. Numerous works have examined the ability of recurrent models [58, 54, 66, 6, 25, 62] and Transformers [17, 65, 62] to model Dyck languages. More recently, significant effort [19, 60, 4] has been devoted to investigating how Transformers can learn to implement learning algorithms in their forward pass to understand the in-context learning phenomenon. Bhattamishra et al. [8] empirically observed that attention-free architectures struggle to implement the nearest neighbor algorithm in comparison to Transformers. Our result takes a step toward understanding this phenomenon by showing that nearest neighbors can be implemented by small-sized Transformers but not by recurrent architectures.

## 2 Definitions

We consider two types of models: Transformers [59] and recurrent models. We use recurrent models to refer more generally to nonlinear RNNs such as LSTMs [26], state-space models [22, 21] as well as variants of linear Transformer [31, 56] which can process inputs in a recurrent manner [31].

For some finite alphabet $\Sigma$, a sequence model is given a sequence in $\Sigma^N$ and depending on the task outputs either $\{0, 1\}$ or a sequence of outputs. Each $s_i \in \Sigma$ from a sequence $s_1 \cdots s_N \in \Sigma^N$ is mapped to a vector in $\mathbf{R}^d$ via an embedding map $\phi : \Sigma \to \mathbf{R}^d$. For Transformers, the input embedding function further takes the position $i$ as input, along with $s_i$. Each layer for Transformers and recurrent models maps inputs from $\mathbb{R}^{N \times d} \to \mathbb{R}^{N \times d}$. A model $M$ has *fixed precision* $p$ if all the parameters of the model as well as the values in the intermediate vectors can be implemented with $p$-bit precision numbers (cf. Appendix B.2).

**Transformers.** Each layer of a Transformer has an attention block followed by an MLP block. The attention block takes as input $\mathbf{X} \in \mathbb{R}^{N \times d}$ and applies the operation $\mathrm{Att}(\mathbf{X}) = \mathrm{softmax}(\mathbf{X}\mathbf{W}_Q^\top \mathbf{W}_K \mathbf{X}^\top)\mathbf{X}\mathbf{W}_V^\top$ where $\mathbf{W}_Q, \mathbf{W}_K, \mathbf{W}_V \in \mathbb{R}^{m \times d}$. For simplicity, we will use $Q(\mathbf{x}_i)$ (and likewise $K(\mathbf{x}_i)$ and $V(\mathbf{x}_i)$) to denote $\mathbf{W}_Q\mathbf{x}_i$. The *width* of the Transformer is $\max(m, d)$, where $m \times d$ is the shape of the projection matrices $\mathbf{W}_Q, \mathbf{W}_K$. For any matrix $\mathbf{A} \in \mathbb{R}^{N \times M}$, the softmax operator is applied row-wise as follows $\mathrm{softmax}(\mathbf{A})_{i,j} = \dfrac{\exp(\mathbf{A}_{i,j})}{\sum_{k=1}^{M} \exp(\mathbf{A}_{i,k})}$. Multi-head attention with $H$ heads is defined as $\text{M-Att}_H(\mathbf{X}) = [\mathrm{Att}_1(\mathbf{X}), \dots, \mathrm{Att}_H(\mathbf{X})]\mathbf{W}_O$ where each $\mathrm{Att}_i(\mathbf{X})$ has its own set of parameters. The matrix $\mathbf{W}_O \in \mathbb{R}^{mH \times d}$ projects the concatenated vector to a vector of dimension $d$. For an input $\mathbf{X} \in \mathbb{R}^{N \times d}$, the output of a layer of Transformer will be $\psi(\text{M-Att}(\mathbf{X})) \in \mathbb{R}^{N \times d}$ where $\psi : \mathbb{R}^d \to \mathbb{R}^d$ is a feedforward network. We use $\text{TF}_{m,p,H}^L$ to denote the class of all Transformers operating over $p$-bit precision numbers with width $m$, $H$ heads, and at most $L$ layers.

**Recurrent Models.** A general recurrent neural network (RNN) takes as input the sequence $\mathbf{x}_1, \dots, \mathbf{x}_N$ where $\mathbf{x}_i \in \mathbb{R}^d$ and produces an output sequence $y_1, \dots, y_N$; in this paper we will mostly consider the case when $y_i \in \{0, 1\}$. An RNN with a hidden state of size $m$ over $p$-bit numbers can be defined as follows. The hidden state is an $mp$-bit memory $\mathbf{h}_i \in \{0, 1\}^{mp}$ and for some $\mathbf{h}_0 \in \{0, 1\}^{mp}$, the RNN computes $\mathbf{h}_t = g_{(t)}(\mathbf{x}_t, \mathbf{h}_{t-1})$ and $y_t = f_{(t)}(\mathbf{h}_t)$ for $t = 1, \dots, N$, and $g_{(t)}$ and $f_{(t)}$ are arbitrary functions. Since the transition function is allowed to be arbitrary, this definition captures the general family of *multi-layer* recurrent architectures including LSTMs, state-space models, and linear Transformers, each of which differs in the way the transition function is defined. For a recurrent model, we say that the *representation size* of a hidden state is $mp$, and its *width* is $m$, i.e. the hidden state consists of $m$ units each of which uses $p$-bits. Throughout the paper, we will use recurrent models or RNNs to refer to the general family of recurrent architectures mentioned above.

By the *representation size* of a model, we will refer to the total number of bits required to represent the model including all the parameters and embeddings. For Transformers, this is $\Theta(mdpH)$. For a recurrent model, the representation size is at least $mp$.

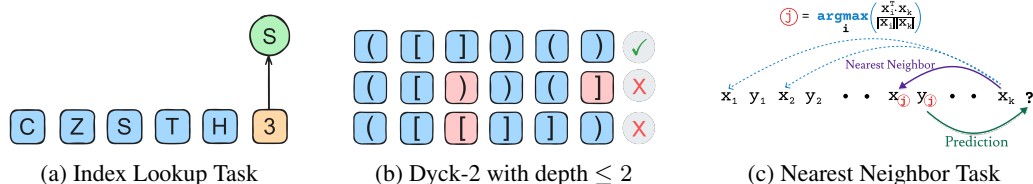

| (a) Index Lookup Task | (b) Dyck-2 with depth $\leq 2$ | (c) Nearest Neighbor Task |

Figure 1: Illustration of a few key tasks considered in our work.

## 3 Index Lookup Task

**Task Description.** We introduce a simple task called Index Lookup (IdxL). In this task a model receives a sequence of tokens $s_1, \ldots, s_N$ (possibly with repetitions) followed by an index $p$ where $p \in [N]$ and the goal of the model is to output the token $s_p$. Here the symbols $s_i$ belong to a finite alphabet $\Sigma$.

This simple and natural task helps illustrate the key tools and building blocks we use to obtain other more general results in this paper: On the one hand, we show how a one-layer Transformer with width $O(\log N)$ can perform this task; on the other hand, we use communication complexity arguments to show that any type of recurrent or state-space model performing this task needs a hidden state with representation size $\Omega(N)$.

Our first result shows that, for any length $N \in \mathbb{N}$, there is a 1-layer Transformer with width $O(\log N)$ that performs the Index Lookup task for all input sequences of length at most $N$. Naïvely one could construct such a transformer by using one-hot encodings as positional embeddings, as they are orthogonal and would allow to attend to the desired index. However, this would require the embedding dimension, and hence the width of the model, to be $\Omega(N)$. Key to our constructions of a width $O(\log N)$ Transformer, both here and in other sections, is a result (Lemma 2 in the Appendix) which states that, in $k = O(\log N/\gamma^2)$ dimensional space we can find $N$ nearly orthogonal vectors. We use such vectors in our construction of the Transformers to allow it to attend almost exactly over desired positions.

**Theorem 1.** *For all $N \in \mathbb{N}$, there is a 1-layer Transformer with width $m = O(\log N)$ and precision $p = O(\log N)$ which performs the index lookup task for all input sequences of lengths up to $N$.*

*Proof Sketch.* For an input sequence $(s_1, \ldots, s_N, p)$, the Transformer uses the embeddings of the position token $p$ and the positional embeddings of the first $N$ inputs to attend over $s_p$, so that the feedforward network can extract the label from the output of the attention block. Our key idea is to use the $N$ almost orthogonal vectors provided by Lemma 2, both as positional embeddings and also as a way to embed the numbers $\{1, \ldots, N\}$, any of which can be used as the index $p$. Formally, let $\mathcal{T}(1), \ldots, \mathcal{T}(N)$ be $N$ vectors of dimension $k = O(\log N)$ such that $\langle \mathcal{T}(i), \mathcal{T}(j) \rangle \leq 1/4$ for $i \neq j$ and $\langle \mathcal{T}(i), \mathcal{T}(j) \rangle \geq 3/4$ for $i = j$.

Formal details of the construction are in Appendix C.2; we provide a sketch. The embedding of each input token is of size $\log |\Sigma| + 2k$ where $\log |\Sigma| + k$ entries are used for the token embeddings and the last $k$ entries are used for the positional embeddings. The query and key matrices are designed so that the query vector $Q(\mathbf{p}) = \eta[\mathcal{T}(p)]$, the key vectors $K(\mathbf{x}_i) = [\mathcal{T}(i)]$ and $K(\mathbf{p}) = [\mathbf{0}_k]$. The value vectors simply contain the token embeddings $V(\mathbf{x}_i) = [\rho(s_i)]$, where $\rho : \Sigma \to \{0,1\}^{|\Sigma|}$ is some binary encoding of $\Sigma$. With such query and key vectors, the dot products in attention, $\langle Q(\mathbf{p}), K(\mathbf{x}_i) \rangle$, are $\geq 3\eta/4$ if $i = p$, and $\leq \eta/4$ otherwise. The dot product of the query vector with itself will be $\langle Q(\mathbf{p}), K(\mathbf{p}) \rangle = 0$. We choose $\eta > 0$ to scale the dot products to amplify the difference between the *high* and *low* dot products. Thus we have,

$$\text{softmax}(Q(\mathbf{p})^\top K(\mathbf{X})) = \frac{\exp(Q(\mathbf{p})^\top K(\mathbf{x}_p))}{\exp(Q(\mathbf{p})^\top K(\mathbf{x}_p)) + \sum_{j \neq p} \exp(Q(\mathbf{p})^\top K(\mathbf{x}_j))} \geq \frac{\exp(\frac{3\eta}{4})}{\exp(\frac{3\eta}{4}) + N \exp(\frac{\eta}{4})}$$

which is at least $\frac{3}{4}$ for some $\eta = \Theta(\log N)$; the total attention weight over the remaining tokens is at most $< \frac{1}{4}$. Recall that the value vectors contain the binary encodings of the input symbols, $V(\mathbf{x}_i) = [\rho(s_i)]$. The attention-weighted average value vector aligns closely with $\rho(s_p)$ as 3/4

weight is on it. It is then straightforward to design a $\mathrm{ReLU}$-FFN that can act as a threshold function to retrieve $\rho(s_p)$ from it, which leads to the desired output. $\qquad\square$

We use results from communication complexity to show that RNNs require essentially $\Omega(N)$ width to solve this and several other problems. In communication complexity, there are two parties, typically called Alice and Bob, each of whom has part of the (discrete) input, and their goal is to compute a function of the combined input using as little communication as possible. Our first key insight here is that the output of an RNN can be computed with a bounded amount of communication when Alice has a prefix of the input and Bob has the remaining part. The resulting protocol will be one-way (Alice to Bob) and one-round. We first state a more general result and then discuss the implications for the Index Lookup problem.

**Theorem 2.** *If an RNN with a hidden state of representation size $mp$ computes any function $f : \Sigma^N \to \{0,1\}$, then for any $K < N$, if Alice has access to $s_1, \ldots, s_K$ and Bob has access to $s_{K+1}, \ldots, s_N$, then there exists a one-way communication protocol with $mp$ bits from Alice to Bob, by which Bob can compute the output of the function $f(s_1 \ldots s_N)$.*

*Proof.* Assume that both Alice and Bob have access to the RNN that represents the function $f$. Alice can provide the sequence $s_1, \ldots, s_K$ to the recurrent model and iteratively update the hidden state from the initial state $\mathbf{h}_0$ to obtain the $K$th hidden state $\mathbf{h}_K$. Alice can then send the hidden state to Bob which requires $mp$ bits. Bob can then update the hidden state using $s_{K+1}, \ldots, s_N$ to obtain $\mathbf{h}_N$, from which he can obtain the output of the RNN. Note that Alice and Bob can compute the output using one-way communication of $mp$ bits. $\qquad\square$

Problems similar to Index Lookup are well-studied in communication complexity; specifically, the INDEX problem (See Appendix B.1) has a one-way communication complexity of $\Omega(N)$ (Fact 3). We deduce a lower bound on the size of the hidden state of RNNs by showing that any RNN that can represent the Index Lookup task can also compute the INDEX problem and since that implies the existence of a one-way communication protocol with $mp$ bits (Theorem 2), it follows that the width of the hidden state $m$ must be $\Omega(N/p)$ (cf. Appendix C.1).

**Theorem 3.** *Any recurrent model with a hidden state of width $m$ using $p$-bits of precision that computes the Index Lookup task for all sequences of length $N$ must have $m \geq N/p$.*

**Discussion.** The above results theoretically formalize intuitive differences between the way Transformers and recurrent models process sequences. Since Transformers have access to $N$ input vectors during their computation, a small-sized attention block can attend over the desired input vector to make the correct prediction. On the other hand, any recurrent model—even with arbitrary positional embeddings—must store all the required information in its hidden state, which lower bounds the size of such models to compute the right output. These intuitions are made rigorous by showing (i) how soft-attention can do lookup using the almost orthogonal vectors, and (ii) small-width RNNs yield a short one-way communication protocol. These lower bounds also apply to causal forms of linear attention architectures where softmax is removed and attention weights become dot products [31].

At first glance, it might seem unfair to compare Transformers and RNNs with the same number of parameters: Transformers have access to $N$ input vectors, whereas RNNs have a fixed-size hidden state. But note that, in practice, empirical research on language models typically compares models of the same size e.g., a 7B Transformer vs a 7B state-space model. Hence, it is natural to ask if Transformers of a particular size can express something that recurrent models cannot.

## 4 Lower Bounds for RNNs and 1-layer Transformers

Whereas Section 3 established a case where one-layer Transformers can be more powerful than RNNs, we next exhibit an example of the opposite phenomenon. Here, the key tool will again be a communication complexity argument, but this time it applies to one-layer Transformers: We establish a communication protocol by which Alice and Bob can compute the output of a one-layer Transformer by exchanging a number of bits that is bounded by the representation size of the Transformer and an overhead that is logarithmic in the input length. The key property here is that this protocol works not just when Alice and Bob have access to a prefix and suffix of a string, but instead works for an *arbitrary* partitioning of the input string (proof is in Appendix D):

**Theorem 4.** *Consider a one-layer Transformer $f \in \text{TF}^1_{m,p,H}$ operating over inputs of length $N$. Consider any disjoint subsets $S_A \cup S_B = \{1, \dots, N\}$, $S_A \cap S_B = \emptyset$. Assume Alice has access to $s_i$ for $i \in S_A$, and Bob has access to $s_i$ for $i \in S_B$. Then Alice and Bob can communicate $3m(p + \log N)H$ bits to compute the output $f(s_1 \dots s_N)$.*

The proof idea is that Alice and Bob first compute their parts of the numerator and denominator of the softmax and exchange these to compute the overall attention output. A naïve implementation of this idea runs into the issue that the exponentiation of logits may exceed the bounds of $p$-bit precision; we circumvent this by first communicating the maximum logit and subtracting it from each logit, keeping the exponentials bounded without altering the resulting attention weights. Theorem 4 is a slightly more general and formal version of a result in Sanford et al. [51, Theorem. 7].

### 4.1 Separation on Bounded Hierarchical Languages

We now use the communication protocol for one-layer Transformers to establish a separation between these and RNNs on bounded Dyck languages. Dyck languages are of central importance in formal language theory, as any context-free language can be expressed in terms of Dyck languages [14]. Due to the boundedness of human memory [41], natural language tends to have more bounded levels of embedding [30, 10]. This has motivated the study of bounded-depth Dyck languages as plausible simple models of the hierarchical structure underlying language [25, 65, 6, 62].

**Task.** Formally, Dyck-(n, k) (cf. Appendix E) is the language of well-matched strings over $n$ types of parenthesis pairs $(_1, )_1, (_2, )_2, \dots, (_n, )_n$, where any prefix has at most $k$ opening parentheses not yet closed. For instance, the string '( [ ] ) ( )' has a maximum depth 2 corresponding to the prefix '( ['. Dyck-(n, k) can be recognized with access to a bounded stack that never holds more than $k$ elements. In fact, each Dyck-(n, k) is a regular language and is accepted by a finite automaton.

We show that there is a linear communication complexity lower bound for Dyck-(n, k), already at $n = k = 2$. However, unlike the communication bound we used in Theorem 3, Alice and Bob now have access not to two halves of the input; rather, Alice and Bob have access to the even and odd positions in the string, respectively. Intuitively, in such a situation, Alice needs to know almost all of the bits available to Bob in order to decide whether a given string is well-bracketed–and vice versa. More formally, they need to exchange at least $N - 1$ bits to decide membership in Dyck-(2, 2):

**Lemma 1.** *Suppose Alice and Bob have the symbols in the odd and even indices of a string $s \in \Sigma^N$ respectively. To each compute whether $s \in$ Dyck-(2, 2), they must exchange at least $N - 1$ bits.*

The proof of Lemma 1 is in Appendix E and is based on fooling sets which is a standard technique to prove lower bounds on communication complexity. Combining Lemmas 4 and 1 entails a lower bound on the width of a Transformer for computing Dyck-(2, 2):

**Theorem 5.** *Consider a one-layer Transformer $f \in \text{TF}^1_{m,p,H}$ deciding membership in Dyck-(2, 2). Then $mH \geq \frac{N-1}{3(p+\log N)}$.*

This result establishes a second separation between one-layer Transformers and RNNs, but now in the other direction: Bounded-depth Dyck languages are regular, and previous works have shown that constant width RNNs can recognize them with standard activation functions such as Sigmoid [25] and ReLU [6]. We further note that two-layer Transformers of sublinear size can model bounded-depth Dyck languages [65].

### 4.2 Lower Bounds on Boolean Functions

There are some notable differences between the types of communication complexity lower bounds for one-layer Transformers (Theorem 4) and for RNNs (Theorem 2). RNNs computing $f$ yield a one-way protocol for contiguous partitions of the input; thus showing a *one way* communication lower bound for such partitions is sufficient to obtain lower bounds on the size of RNNs. Transformers computing $f$ yield a multi-way protocol that work for *arbitrary partitions* of the input; thus showing a lower bound for *any* partition is sufficient to establish lower bounds on the size of the Transformer. For Dyck-(2, 2), contiguous partitions are not a hard case, and in fact, communicating $\leq 2$ open brackets from the first half is sufficient. This is why the lower bound of Lemma 1 does not apply to RNNs.

**Lower Bounds.** Despite the differences discussed above, there are several Boolean functions, for which we can establish that when widths are bounded, neither one-layer Transformers nor RNNs

can compute them. The Equality function $\text{EQ} : \{0,1\}^N \to \{0,1\}$ is a Boolean function defined as $\text{EQ}(\mathbf{x}) = \mathbb{I}[(\mathbf{x}_1, \ldots, \mathbf{x}_{N/2}) = (\mathbf{x}_{N/2+1}, \ldots, \mathbf{x}_N)]$. A related problem is *Disjointness*: given two vectors $\mathbf{x}, \mathbf{y} \in \{0,1\}^{N/2}$, the function $\text{DISJ}(\mathbf{x}, \mathbf{y}) = \max \mathbf{x}_i \mathbf{y}_i = \mathbb{I}[\mathbf{x}^T \mathbf{y} > 0]$. Both the functions Equality and Disjointness are known to have communication complexity $\Omega(N)$ (see Appendix B.1) and Theorems 4 and 2 imply that both one-layer Transformers and RNNs must have width $\Omega(N)$ to represent them. In the next section, we show that these lower bounds do not apply to two-layer Transformers. Additionally, it is worth noting that the functions EQ and DISJ can also be expressed in the form of 2-CNFs with $O(N)$ terms. Hence, a more general consequence of the limitations of RNNs (and one-layer Transformers) is that with width $o(N)$, they cannot compute certain functions in the class of uniform $\text{AC}^0$.[3] It is interesting since the class of uniform $\text{AC}^0$ is considered one of the simplest classes of Boolean circuits and even the expressive power of hard-attention Transformers has been shown to be within this class [24]. In Appendix F.1, we provide an alternate proof of the lower bound for RNNs computing Equality based on their relation to DFAs.

## 5 Representational Capabilities of 2-layer Transformers

In Section 4, we showed that single-layer Transformers and recurrent models must have size *linear* in the input length to express natural Boolean functions such as EQ. In this section, we show that two-layer transformers overcome these limitations by efficiently expressing such Boolean functions and more general forms of associative recall tasks, such as simulating the nearest neighbor algorithm.

### 5.1 Representing Boolean Functions

We start by showing that two-layer Transformers of poly-logarithmic size can express the Equality function (proof is in Appendix F.2). The input domain need not necessarily be the Boolean vectors $\{0,1\}^N$; rather, the construction works for sequences over any finite alphabet $\Sigma$.

**Theorem 6.** *For any $N \in \mathbb{N}$, there exists a 2-layer Transformer $f \in \text{TF}^2_{m,p,2}$ where width $m = O(\log N)$ and precision $p = O(\log N)$ such that $f(\mathbf{x}) = \text{EQ}(\mathbf{x})$ for all $\mathbf{x} \in \{0,1\}^N$.*

The construction is based on tools developed in Section 3. The broad idea is as follows. In the first layer, at each position $i > N/2$, an attention head attends to position $i - N/2$ and copies the input $x_{i-N/2}$. A feedforward network then checks whether the retrieved value is equal to $x_i$. The second layer simply uses uniform attention over the outputs of the previous layer to check if there is a mismatch at any position. Importantly, we show that the above strategy can be implemented with a representation size $O((\log N)^3)$.

Generalizing this result, we find that two-layer Transformers with logarithmic width can express a more general class of Boolean functions: thresholds of $k$-sparse features, a class including functions such as Equality and Disjointness. Since such functions cannot be expressed by one-layer Transformers and recurrent models with width $o(N)$, these results imply a *separation* on Equality and Disjointness: these functions can be expressed by small-sized two-layer Transformers whereas one-layer Transformers and recurrent models must grow linearly with input length to represent them.

### 5.2 Implementing the Nearest Neighbors Algorithm

The goal of the nearest neighbor task (NSTNB) is to analyze whether a sequence modeling architecture can implement the well-known nearest neighbor algorithm to make predictions. Our description follows closely to the one used by Bhattamishra et al. [8] for their experiments.

**Nearest Neighbors.** In the NSTNB task, a model is provided with a sequence of vectors and labels $(\mathbf{x}_1, y_1, \ldots, \mathbf{x}_{k-1}, y_{k-1}, \mathbf{x}_k)$ where $N/2 < k \leq N$, the input *unit* vectors $\mathbf{x}_i \in \mathbb{R}^d$ and labels $y_i \in \{0,1\}$. For each $\mathbf{x}_k$ where $k > N/2$, the output is the label corresponding to the nearest neighbor in $(\mathbf{x}_1, \ldots, \mathbf{x}_{k-1})$, that is, if $j = \arg\max_{i \in [k-1]} \mathbf{x}_k^\top \mathbf{x}_i$ or $j = \arg\min_{i \in [k-1]} \|\mathbf{x}_k - \mathbf{x}_i\|_2$, then the output for $\mathbf{x}_k$ is the label $y_j$. Since we are working with unit vectors, maximizing the inner product is equivalent to minimizing the $\ell_2$ distance. If the second half of the sequence $\mathbf{x}_{\frac{N}{2}+1}, \ldots, \mathbf{x}_N$ is a permutation of the first half $\mathbf{x}_1, \ldots, \mathbf{x}_{\frac{N}{2}}$ then the task reduces to the Multi-Query Associative Recall (MQAR) task [2] (cf. Appendix G).

---

[3]The class $\text{AC}^0$ contains polynomial size AND/OR circuits with unbounded fan-in and constant depth.

**Assumptions.** We will make two assumptions about the problem. The first assumption is that all input vectors are of unit norm, i.e., $\|x\|_2 = 1$ and the second is the existence of a margin between the dot product with the nearest neighbor and the dot product with other input vectors, i.e. there exists $\gamma \geq N^{-c}$ for some universal constant $c$, such that for any $N/2 < k \leq N$, if $j^* = \arg\max_{i \in [k-1]} \mathbf{x}_k^\top \mathbf{x}_i$, then $\mathbf{x}_k^\top \mathbf{x}_{j^*} \geq \mathbf{x}_k^\top \mathbf{x}_i + \gamma$ for any $i \neq j^*$.

The following is one of our main results which states that two-layer Transformers of logarithmic size can implement the nearest neighbor algorithm in their forward pass and as a corollary can also perform associative recall tasks like MQAR (Proofs in Appendix G.2).

**Theorem 7.** *For any $N \in \mathbb{N}$, there exists a 2-layer Transformer $f_{NN} \in \mathrm{TF}_{m,p,2}^2$ with width $m = O(\log N)$ and precision $p = O(\log N)$ such that $f_{NN}$ computes the nearest-neighbor task all sequences of length at most $N$ satisfying the assumptions above.*

The broad idea of the construction is to identify the nearest neighbor input $\mathbf{x}_{j^*}$ and retrieve the position of the corresponding label $y_{j^*}$ in the first layer. The second layer then uses this positional information to retrieve the desired label. There are a few challenges to implementing this strategy which we address in our construction. First, note that for input vectors $\mathbf{x}_1, \ldots, \mathbf{x}_k$, naively using them with dot-product attention will result in the query input $\mathbf{x}_k$ having maximum dot product and hence maximum attention weight over itself. Second, the dot product with some label vectors $y_i$s could be higher than the dot product with the nearest neighbor $\mathbf{x}_{j^*}$. Third, the positional information must be retrieved using soft-attention in a way that it can be used in the next layer to obtain the desired label. Our intuitive, though somewhat involved, construction deals with these issues to ensure that a two-layer Transformer with $O((\log N)^3)$ total size implements the nearest neighbor algorithm.

**Theorem 8.** *Any recurrent model with a hidden state of width $m$ with $p$-bits of precision that can perform the nearest neighbor task for all inputs of length $N$ must have $m \geq N/2p$.*

The lower bound for recurrent models follows via a reduction from the Disjointness problem.

**Discussion.** Prior works [8, 2] have empirically demonstrated that Transformer-based LLMs can exhibit mechanisms such as nearest neighbors and MQAR. Further, on synthetic setups, they have observed that recurrent models struggle to perform these tasks compared to Transformers. Our results take a step towards understanding the differences in the performance between the two architectures.

# 6 Empirical Analysis

While we focus on the differences in the representational capabilities of Transformers and recurrent models, it is natural to examine if differences of a similar nature arise in their empirical performance. One thing to keep in mind is that positive results regarding expressiveness presented in earlier sections do not imply that models can *learn* such tasks. With regard to negative results, they do imply that when the sequence length is much larger than the size of the hidden state or width of the model, then the model will be incapable of representing the task and consequently fail to learn the task. However, even for one-layer recurrent models with hidden states of size 128 with 64 bits of precision, our lower bound applies at lengths over 8k.

In this section, we investigate the performance of Transformers and recurrent models on tasks such as Index Lookup and recognizing bounded Dyck languages on sequences of small lengths ($< 1000$). Our experiments are designed to answer the following questions: (1) Are one-layer Transformers better than larger recurrent models on the Index Lookup task? (2) Are recurrent models and two-layer Transformers better than one-layer Transformers at recognizing the Dyck-(2, 2) language? Importantly, as our results concern the scaling of the model size with the input length, we are specifically interested in the behavior of different models across input lengths.

We also explore the performance of models on string equality in Appendix H.2. Tasks like NSTNB and MQAR have already been analyzed empirically in prior works [8, 2] so we do not include them.

**Setup and Training details.** We train the models with cross-entropy loss using the Adam optimizer [32]. The models are trained for up to 250k steps where at each step we sample a fresh batch of 64 training examples – resulting in $\approx 16$ million examples over 250k steps. The models are evaluated on 5000 examples for each task. For each model, we tune the various hyperparameters, notably across learning rates $\in \{1e\text{-}2, 5e\text{-}3, \ldots, 1e\text{-}6\}$ to find the best-performing model. The details of the data generation method, hyperparameters, and implementation details can be found in Appendix H.

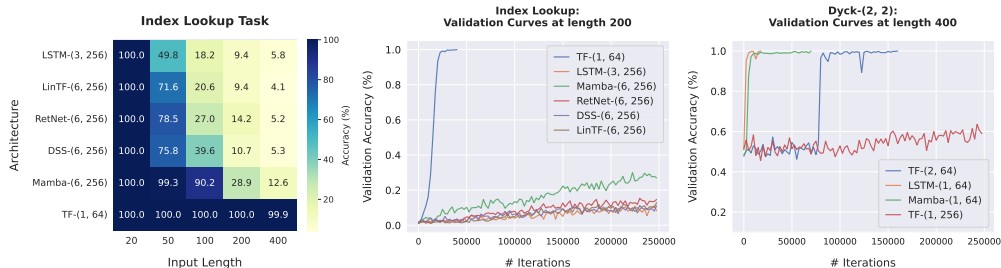

Figure 2: Performance of models on the Index Lookup and bounded Dyck task. Labels such as TF-(1, 64) denote Transformers with 1 layer and 64 widths. See Section 6 for more details.

**Index Lookup Task.** We compare the performance of one-layer Transformers with five different recurrent models – LSTMs [26], state space models such as DSS [22] and Mamba [21], linear Transformers [31], and its variant RetNet [56]. We explore the performance across various lengths $\in \{20, 50, 100, 200, 400\}$. We evaluate relatively small-sized Transformers with widths $64$ against recurrent models with up to 6 layers and widths or hidden state size of $256$. The size of the alphabet in the experiments is $|\Sigma| = 64$. Figure 2 (left) depicts the performance of all models across various lengths and Figure 2 (middle) depicts the validation curves during training on examples of length $200$. As depicted by the figures, while one-layer Transformers with width $64$ achieve near-perfect accuracy within a few thousand steps, the performance of relatively larger recurrent or state-space models degrades on lengths over $100$ and they fail to learn even with $10\times$ training iterations. We explore the influence of width on the performance of Mamba in Appendix H.1.

**Bounded Dycks.** For Dyck-2 with depth at most 2, our separation results apply to one-layer Transformers and nonlinear recurrent models such as LSTMs but not to linear RNNs such as state-space models and linear Transformers. Hence, we are primarily interested in the difference in performance between one-layer Transformers and LSTMs. In our experiments, we compare the performance of one-layer Transformers with relatively smaller recurrent models such as LSTMs and two-layer Transformers. We also include Mamba for reference. We consider LSTMs and Mamba with hidden state sizes of 64 and similarly, two-layer Transformers with width 64. We evaluate a one-layer Transformer with a width of 256 across lengths $\in \{20, \dots, 400\}$ most of which are smaller than the width of the model. We observe that one-layer Transformers achieve near-perfect accuracy up to lengths $100$ but struggle on higher lengths. In contrast, small-sized recurrent models as well as two-layer Transformers can achieve near-perfect accuracy for lengths up to $400$. Figure 2 (right) depicts the validation curve of the models on examples of length $400$.

# 7 Discussion and Final Remarks

Based on prior theoretical results, it is known that, while recurrent models can express any regular language [34, 33], Transformers with logarithmic precision can only express languages in the class of uniform constant depth threshold circuits (TC$^0$) [40]. These results indicate that—under standard conjectures—Transformers are unable to represent certain state-tracking tasks that recurrent models can represent. With such results, it might appear that Transformers are less expressive than recurrent models–potentially at odds with the persistent practical success of Transformer-based LLMs. Our findings, however, show that when the model size is constrained relative to the sequence length, a variety of tasks relevant to practice can be represented by small-sized Transformers but not by recurrent models. Our results suggest that the attention mechanism does lead to expressiveness that cannot be replicated by recurrent architectures even with arbitrary transition functions.

**Limitations.** A general limitation of this line of work is that positive expressivity results do not imply that the problems under consideration are learnable. Additionally, while lower bounds for an architecture imply difficulty in learning, when using double precision these results only apply to very long sequences in practice. Our results (and probably techniques) do not imply any limitations on two-layer Transformers; this is left as an open question. We note that communication complexity-based techniques akin to Theorem 4 cannot exist for two-layer Transformers (cf. Appendix F.4). Hence, we believe that other tools will be needed to prove lower bounds for two-layer Transformers.

## Acknowledgements

We thank Dibyayoti Dhananjay Jena, Arkil Patel, and Charles London for insightful and helpful discussions on this work. We also thank Clayton Sanford, Will Merrill, and the anonymous reviewers for their valuable feedback and constructive suggestions.

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

# Contents

## A Clarifications

**(1)** *What do the results presented in the paper imply about the learnability of the tasks considered?*

The lower bounds on recurrent models and one-layer Transformers have negative implications for learnability but the positive results do not have any nontrivial implications. The lower bounds on the size of recurrent models and one-layer Transformers imply that unless the width of the models grows linearly with respect to the input length, they cannot represent and consequently cannot learn such tasks. Note, however, that even though the input length need not be unboundedly long for our lower bounds to apply, they still need to be sufficiently large $N \gtrsim m$ and do not apply at the lengths

considered in our experiments. The results indicating that Transformers can express tasks like Index lookup, string equality or nearest neighbors do *not* imply that they can learn those tasks in practice. The experiments in Sections 6 and H as well as from prior works [8] seem to indicate that small-sized Transformers perform reasonably well on these tasks.

**(2)** *For the upper bounds on the total size of Transformers to express functions, does it include the input and positional embeddings or just the Transformer model with attention and feedforward block?*

Yes, when we say that Transformers with total size $O(f(N))$ or poly-logarithmic in $N$ can express a certain task, it includes the input and positional embeddings. The total representation size indicates the total number of bits required to represent all the parameters of the model. Our results imply that a Transformer with total size $O((\log N)^3)$ can represent tasks such as nearest neighbors, Equality, etc, whereas the size of any recurrent model must be $\Omega(N)$. For our constructions of Transformers, we ensure that the positional embeddings can be generated in log space (discussed in Appendix B) and need not be stored as a $N \times d$ matrix for the required computations. Lastly, it is worth noting that all our lower bounds for recurrent models apply even if they have arbitrary positional embeddings stored in $N \times d$ matrix, and hence the size of recurrent models is $\Omega(N)$ excluding the embeddings.

**(3)** *For the constructions with Transformers, why can't those results follow from some frameworks like RASP?*

RASP [61, 36] is a programming language aiming to abstract computations that transformers can compute. While one might be able to construct RASP programs for some of the tasks we have considered, such as Index Lookup, such constructions would not entail results similar to ours, because RASP substantially abstracts from implementational aspects. For instance, RASP allows MLPs to compute arbitrary functions, and attention is not computed from dot products of keys and queries. It is not clear if general-purpose translations from RASP to realistic transformers would be able to recover our efficient bounds, e.g., logarithmic size.

**(4)** *Do the negative results in experiments imply that those architectures cannot learn those tasks?*

The differences in the performance of models apply more to the rate of learning than a binary form of success/failure. For instance, in the Index Lookup task, one can see that in Figure 2, one-layer Transformers achieve near-perfect accuracy in a few thousand steps whereas recurrent models fail to achieve high accuracy even after being trained for $25\times$ steps. It can still be true that if the models are trained for much longer or are much bigger, they might achieve high accuracy. See Figure 3 which depicts the performance of Mamba across various sizes and lengths. For tasks like Index lookup, we observe that Transformers achieve near-perfect accuracy across several learning rates whereas recurrent architectures fail on all the learning rates we tried. A similar thing is true for other tasks such as Dyck-(2, 2) where one-layer Transformers seem to learn at a much slower rate compared to LSTMs and two-layer Transformers even for the lengths where they do achieve near-perfect accuracy.

**(5)** *Do the lower bounds for Transformers and RNNs apply with additional components such as layer norm, residual connections, positional embeddings, etc?*

Yes, for our lower bounds for one-layer Transformers and Recurrent models, they still apply if the models have additional components including arbitrary positional embeddings, layer norms, etc. We only assume that the attention mechanism operates over finite precision and the hidden state of RNN is in finite precision. The results do not make any assumptions about the computational limits of the remaining components. They apply even when the output of the attention block or the hidden state of an RNN is processed by an *arbitrary function*.

# B   Preliminaries

## B.1   Communication Complexity

Our lower bounds for recurrent models and 1-layer Transformers are based on communication-complexity bounds. We assume some familiarity with communication complexity (see Rao and Yehudayoff [48], Kushilevitz and Nisan [35] for an introduction). We will primarily focus on the two-party setting where the communication complexity of a function indicates the number of bits two parties must exchange in order to compute the output of a function. If Alice and Bob want to compute a function $f : \{0, 1\}^N \to \{0, 1\}$ where Alice has the bits in the indices $I \subset [N]$ and Bob has the bits in indices $[N] \setminus I$, the communication complexity of $f$ over that partition is the minimum number

of bits they must exchange to compute the output for all inputs $x \in \{0, 1\}^N$. Alice and Bob are allowed unbounded computational power. If Alice and Bob must exchange at least $k$ bits to compute a function $f(x) \forall x \in \{0, 1\}^n$ over any partition of the input then we say that the communication complexity $C(f) \geq k$. If they use a communication protocol where only one party is allowed to send bits to the other party, then it is called one-way communication, and the communication complexity of the function in that setting is referred to as one-way communication complexity.

For our results, we will use the following three well-known facts about the communication complexity of the Disjointness, Equality, and the INDEX problem.

**Disjointness.** The disjointness function takes two sets $A, B \subseteq [N]$ as input and returns 0 if the two sets are disjoint and returns 1 otherwise. This can also be seen as a function $\text{DISJ} : \{0, 1\}^N \times \{0, 1\}^N \to \{0, 1\}$ over two Boolean inputs such that $\text{DISJ}(a, b) = \max a_i b_i = \mathbb{I}[a^T b > 0]$. If Alice has the input vector $a \in \{0, 1\}^N$ and Bob has the vector $b \in \{0, 1\}^N$, then the communication complexity of DISJ indicates the minimum number of bits that Alice and Bob will have to exchange to determine the output of the DISJ function. The following is a well-known fact about the disjointness problem,

**Fact 1.** *(Disjointness [64]) If Alice and Bob have two inputs $a, b \in \{0, 1\}^N$, then any deterministic communication protocol used by them to compute $\text{DISJ}(a, b) = \max a_i b_i$ must exchange at least $N$-bits. Moreover, the randomized communication complexity of the DISJ is $\Omega(N)$.*

**Fact 2.** *(Equality [48, Ch. 1]) If Alice and Bob have two inputs $a, b \in \{0, 1\}^N$, then any deterministic communication protocol used by them to compute $\text{EQ}(a, b) = \mathbb{I}[a = b]$ must exchange at least $N$-bits.*

**Fact 3.** *(INDEX [27]) If Alice and Bob have two inputs $a \in \{0, 1\}^N$ and $b \in [N]$ respectively, then any deterministic communication protocol used by Alice to send bits to Bob must require $N$ bits for Bob to compute $\text{INDEX}(a, b) = a_b$. Moreover, the one-way randomized communication complexity of the INDEX is $\Omega(N)$.*

One thing to note about the Equality problem is that although the deterministic communication complexity of the EQ problem is $\Omega(N)$, the randomized communication complexity is $O(\log N)$. For the Disjointness and INDEX problems, the randomized communication complexity is $\Omega(N)$ as well. In other words, even if the two parties are allowed to compute the output of the function correctly with high probability (say $> 2/3$), even then the number of bits they must exchange is $\Omega(N)$.

## B.2 Finite Precision Implementation

In this work, we are interested in the expressiveness of finite precision models. For our constructions with Transformers, we will work with $p$-bit numbers where $p = \Theta(\log N)$ where $N$ is the maximum length of the input string.

In particular, for some sufficiently large constant $K_c > 0$, we will work with numbers between $-N^{K_c}$ to $N^{K_c}$ with a step size of $\Delta = \frac{1}{N^{K_c}}$. If $\mathbb{Q}_p$ is the set of all such numbers then $\mathbb{Q}_p = \{-N^{K_c}, -N^{K_c} + \Delta, \ldots, 0, \Delta, 2\Delta, \ldots, N^{K_c}\}$. Hence, the size of the set is $|\mathbb{Q}_p| = N^{2K_c} \implies p = 2K_c \log N = \Theta(\log N)$. For any real number $z \in \mathbb{R}$, in the finite precision implementation, $z$ is rounded down to the nearest $\hat{z}$ such that $\hat{z} N^{K_c} \in \mathbb{Z}$. If $z > N^{K_c}$, then $z$ is rounded down to $N^{K_c}$. In our constructions with Transformers, all the parameters and intermediate values follow the above implementation.

## B.3 Technical Tools

For our constructions of Transformers, we will use the following result about high dimensional vectors. The statement essentially says that at a high dimension $k$ the number of vectors that are almost orthogonal to each other is exponential in $k$ even though the number of orthogonal vectors can be at most $k$.

**Lemma 2.** *For any $N$, there exists $N$ $k$-dimensional vectors $\mathcal{T}_1, \ldots, \mathcal{T}_N$ where $k = O(\frac{1}{\gamma^2} \log N)$ and each entry of the vectors is in $\{-\frac{1}{\sqrt{k}}, \frac{1}{\sqrt{k}}\}$ such that*

$$\langle \mathcal{T}_i, \mathcal{T}_j \rangle \begin{cases} \geq 1 - \gamma & \text{if } i = j, \\ \leq \gamma & \text{otherwise.} \end{cases}$$

It is quite straightforward to see why this is true. It follows from a simple application of the probabilistic method. Suppose, one samples $N$ $k$-dimensional vectors $\mathbf{X}_1, \ldots, \mathbf{X}_N$ independently at random such that each entry of each vector is drawn uniformly from $\{-\frac{1}{\sqrt{k}}, \frac{1}{\sqrt{k}}\}$. Hence, each vector $\mathbf{X}_i \in \{-\frac{1}{\sqrt{k}}, \frac{1}{\sqrt{k}}\}^k$ and the expectation of the dot product of any two vectors $\mathbb{E}[\langle \mathbf{X}_i, \mathbf{X}_j \rangle] = 0$ for all $i \neq j$. Since the dot products of any two vectors $\langle \mathbf{X}_i, \mathbf{X}_j \rangle$ is a bounded random variable, we can apply Hoeffding's inequality to get

$$\mathbb{P}[|\langle \mathbf{X}_i, \mathbf{X}_j \rangle| \geq \gamma] \leq 2 \exp\left(\frac{-k\gamma^2}{4}\right).$$

Taking a union bound over at most $N^2$ pairs of dot products and setting $2N^2 \exp\left(\frac{-k\gamma^2}{4}\right) < 1/2$, we get that for $k = \frac{8}{\gamma^2} \log 2N$, the probability of all dot products $\langle \mathbf{X}_i, \mathbf{X}_j \rangle$ being less than $\gamma$ at the same time is at least $1/2$. Since the probability of the event is nonzero, it follows that the statement in the lemma is true.

In our construction, we will use such vectors as positional embedding vectors. While Lemma 2 implies the existence of vectors, storing $N$ such vectors will require $\Theta(N)$ space. We would like to generate $i$-th vector $\mathcal{T}_i$ when necessary in polynomial time using log space without storing all of them together. To achieve that, we use a derandomization of the Johnson-Lindenstrauss (JL) Lemma [29] that can output the $i$-th vector in log-space and polynomial time [53].

We use a slightly modified version of the JL lemma which preserves inner products over unit vectors.

**Lemma 3.** *(Inner product preservation [1]) Let $\epsilon \in (0, 1/2)$ and let $\mathcal{Q} \subset S^{d-1}$ be a set of $N$ unit norm vectors of dimension $d$. For $k = \frac{c_2 \log N}{\epsilon^2}$, there exists a linear map $\mathcal{T}(x) = \frac{1}{\sqrt{k}} A x$ where each entry of $A : \mathbb{R}^d \to \mathbb{R}^k$ is in $\{-1, 1\}$ such that for all $u, v \in \mathcal{Q}$,*

$$|\langle u, v \rangle - \langle \mathcal{T}(u), \mathcal{T}(v) \rangle| \leq \epsilon$$

The above result has interesting implications which we will use in our constructions. Let $\mathbf{x}_1, \ldots, \mathbf{x}_N \in \mathbb{R}^N$ be $N$ unit norm vectors that form the basis of $\mathbb{R}^N$ which implies $\langle \mathbf{x}_i, \mathbf{x}_j \rangle = 1$ if $i = j$ and is 0 otherwise. Then there exists a map $\mathcal{T}$ such that the vectors $\mathcal{T}(\mathbf{x}_1), \ldots, \mathcal{T}(\mathbf{x}_N) \in \mathbb{R}^k$ have dot products $\langle \mathcal{T}(\mathbf{x}_i), \mathcal{T}(\mathbf{x}_j) \rangle = 1 \pm \epsilon$ if $i = j$ and is $0 \pm \epsilon$ otherwise.

We will use JL transformations of the standard basis vectors $\mathbf{e}_1, \ldots, \mathbf{e}_N$ where $\mathcal{T}(1), \ldots, \mathcal{T}(N)$ will refer to $k = O(\log N)$ dimensional vectors such that their inner product is $\approx 1$ with themselves and is $\approx 0$. Intuitively, we can use such vectors to get a Transformer to attend to unique positions.

**Corollary 8.1.** *For any $N$, there exists $N$ $k$-dimensional vectors $\mathcal{T}(1), \ldots, \mathcal{T}(N)$ where $k = O(\log N)$ and each entry of the vectors is in $\{-\frac{1}{\sqrt{k}}, \frac{1}{\sqrt{k}}\}$ such that*

$$\langle \mathcal{T}(i), \mathcal{T}(j) \rangle \begin{cases} \geq 3/4 & \text{if } i = j, \\ \leq 1/4 & \text{otherwise.} \end{cases}$$

## C  Index Lookup Task

**Index Lookup.** The index lookup task (IdxL) is a multi-class classification task where a model receives a sequence of tokens $x_1, \ldots, x_N$ followed by a position token $p$ where $p \in [N]$ and the goal of the model is to output the token $x_p$ at position $p$. Here the symbols $x_i$ belong to a vocabulary $\Sigma$, a finite set of symbols. The sequence $x = (x_1, \ldots, x_N)$ can have repetitions. More precisely, we say a model computes the function $\text{IdxL} : \Sigma_{\leq N} \times [N] \to \Sigma$ if for all inputs $(x, p)$, the model outputs $\text{IdxL}(x, p)$.

### C.1  Recurrent models must be wide to perform Index Lookup

**Theorem 3.** *Any recurrent model with a hidden state of width $m$ using $p$-bits of precision that computes the Index Lookup task for all sequences of length $N$ must have $m \geq N/p$.*

*Proof.* The proof follows naturally via a reduction from the INDEX problem in communication complexity. Assume the vocabulary size for the IdxL is at least 2, pick any two symbols from the vocabulary, and map them to 0 and 1. Suppose Alice has a sequence $a \in \{0,1\}^N$ and Bob has an index $i \in [N]$. Both of them have access to a recurrent model as described in Section 2, which can perform the IdxL task perfectly. Alice can then provide the sequence $a$ to the recurrent model using any two symbols in the vocabulary $\Sigma$ and iteratively update the hidden state to obtain the $N$th hidden state $h_N$. Alice can then send the hidden state to Bob which requires $mp$ bits. Bob can then provide the position token based on the index $i$ and compute the output to figure out whether $a_i$ is 0 or 1.

Note that Alice and Bob can compute the output using one-way communication of $mp$ bits and hence based on Fact 3, it must be the case that $mp \geq N$.

$\square$

**Associative Recall.** A similar argument can be used to show that any recurrent model that can correctly perform the single query associative recall task [3] must have a width or hidden state of size at least $\Omega(N/p)$. In the associative recall task, a model is presented with a sequence of symbols and labels $x_1, y_1, \ldots, x_N, y_N$ followed by a query symbol $x_q$ which is one of the symbols presented earlier $(x_1, \ldots, x_N)$. The goal of a model is to output the label $y_i$ corresponding to the symbol $x_q = x_i$ where $i = 1, \ldots, N$. In this task, the symbols $(x_1, \ldots, x_N)$ must be distinct.

**Proposition 9.** *Any recurrent model with a hidden state of size $m$ over p-bits of precision that computes the associative recall task for all sequences of length $N$ must have $m \geq N/p$.*

A similar reduction from the INDEX problem can be used to show a lower bound for recurrent models. Both Alice and Bob have the description of the recurrent model and both of them can have a predetermined protocol for the sequence of symbols $x_1, \ldots, x_N$. Alice can use the recurrent model to compute the hidden state $\mathbf{h}_{2N}$ by providing it with inputs $x_1, a_1, \ldots, x_N, a_N$ where the bits in $a\{0,1\}^N$ are provided as labels. Alice can send the hidden state using $mp$ bits and Bob can provide the symbol $x_i$ corresponding to the query index $i$ and compute the output. Hence, the size of the hidden state of the RNN must be at most $N/p$.

### C.2 1-layer Transformer with small width can perform Index Lookup

While any form of recurrent model must have a width or hidden state of size $\Omega(N)$ to perform the index lookup task, we show that a 1-layer Transformer with width $O(\log N)$ can perform the index lookup task.

**Theorem 1.** *For all $N \in \mathbb{N}$, there is a 1-layer Transformer with width $m = O(\log N)$ and precision $p = O(\log N)$ which performs the index lookup task for all input sequences of lengths up to $N$.*

*Proof.* For an input sequence $(s_1, \ldots, s_N, p)$, the Transformer uses the embeddings of the position token $p$ and the positional embeddings of the first $N$ inputs to attend over $s_p$, so that the feedforward network can extract the label from the output of the attention block. Our key idea is to use the $N$ almost orthogonal vectors provided by Lemma 2, both as positional embeddings and also as a way to embed the numbers $\{1, \ldots, N\}$, any of which can be used as the index $p$. Formally, let $\mathcal{T}(1), \ldots, \mathcal{T}(N)$ be $N$ vectors of dimension $k = O(\log N)$ such that $\langle \mathcal{T}(i), \mathcal{T}(j) \rangle \leq 1/4$ for $i \neq j$ and $\langle \mathcal{T}(i), \mathcal{T}(j) \rangle \geq 3/4$ for $i = j$.

Recall that, in the index lookup task, the alphabet $V = \Sigma \cup [N]$ consists of symbols $s_i$ from a set $\Sigma$ and the index tokens $[N] = \{1, \ldots, N\}$. The embeddings of each input token will be of size $\log |\Sigma| + 2k$ where $\log |\Sigma| + k$ indices are reserved for the token embeddings and the last $k$ indices are used for the positional embeddings. Suppose we use a binary encoding for symbols in $\Sigma$, and $\rho : \Sigma \rightarrow \{0,1\}^{\log |\Sigma|}$ is such that $\rho(s)$ is the binary encoding of $s \in \Sigma$. The token embedding of each symbol $s_j \in \Sigma$ is $\rho(s_j)$ of length $\log |\Sigma|$ followed by $k$ zeros. For inputs sequence $s_1, \ldots, s_N$, the embedding vectors will be of the form $\mathbf{x}_i = [\rho(s_i), \mathbf{0}_k, \mathcal{T}(i)]$.

The embedding for any index token $p \in [N]$ contains $\log |\Sigma|$ zeros followed by the vector $\mathcal{T}(p)$. In other words, the embedding for the index token will be identical to the positional embeddings used in the first $N$ tokens. The embedding vector corresponding to the index token $p$ will be of the form $\mathbf{p} = [\mathbf{0}_{\log |\Sigma|}, \mathcal{T}(p), \mathbf{0}_k]$.

The output of the 1-layer Transformer is computed by applying attention over the input sequence with the query vector corresponding to the last token, followed by an application of the feedforward network over the output of the attention block. We can define the matrices $W_Q = \eta[\mathbf{O}; \mathbf{I}; \mathbf{O}]$, $W_K = [\mathbf{O}, \mathbf{O}, \mathbf{I}]$, $W_V = [\mathbf{I}, \mathbf{O}, \mathbf{O}]$, where $\eta > 0$ is a parameter that will be specified later, $\mathbf{I}$ is a square identity matrix and $\mathbf{O}$ is a zero-matrix of the appropriate shape. With these definitions we get that the query vector $Q(\mathbf{p}) = \eta[\mathcal{T}(p)]$ contains the middle part of the input embedding, the key vectors $K(\mathbf{x}_i) = [\mathcal{T}(i)]$ contain the last part of the input embedding and the value vectors $V(\mathbf{x}_i) = [\rho(s_i)]$ contain the first part of the input embedding.

With these query and key vectors, the dot products in attention satisfy:

$$\langle Q(\mathbf{p}), K(\mathbf{x}_i)\rangle \begin{cases} \geq 3\eta/4 & \text{if } i = p, \\ \leq \eta/4 & \text{if } i \neq p \end{cases}.$$

Additionally, the dot product of the query vector with itself will be $\langle Q(\mathbf{p}), K(\mathbf{p})\rangle = 0$.

To retrieve the required token with the $\mathrm{softmax}$ operator, consider

$$\mathrm{softmax}(Q(\mathbf{p}), K(X)^\top)_p = \frac{\exp(\langle Q(\mathbf{p}), K(\mathbf{x}_p)\rangle)}{\exp(\langle Q(\mathbf{p}), K(\mathbf{x}_p)\rangle) + \sum_{j \neq p} \exp(\langle Q(\mathbf{p}), K(\mathbf{x}_j)\rangle)}$$

$$\geq \frac{\exp(\frac{3\eta}{4})}{\exp(\frac{3\eta}{4}) + N \exp(\frac{\eta}{4})}$$

which is $> \frac{3}{4}$ for any $\eta > 2\log(3N)$. That is, for $\eta > 2\log(3N)$, the attention weight over input token $x_p$ with a query token $p$ will be greater than $3/4$, and hence the total attention weight over the remaining tokens will be less than $1/4$.

Recall that the value vectors contain the binary encodings of the input symbols, $V(\mathbf{x}_i) = [\rho(s_i)]$. Let $q_1, \dots, q_N$ be the probabilities assigned to the $N$ tokens by the attention operator. Note that $q_p \geq 3/4$. Let $\bar{z} = \sum_i q_i \rho(s_i)$. Note that if the $j$-th bit of $\rho(s_p)$ is 1, then $\bar{z}_j \geq 3/4$ and otherwise, $\bar{z}_j \leq 1/4$. From there a ReLU-FFN can transform the vector $\bar{z}$ to the vector $\rho(s_p)$ which is the desired output. □

**Discussion.** While we showed that a one-layer Transformer with $O(\log N)$ width can represent the index lookup task, it is unclear whether a one-layer Transformer with a small width can express the associative recall task. The construction above cannot be adapted in a straightforward manner to show that one-layer Transformers can represent the associative recall task. From the results in Section G on nearest neighbors, it follows that two-layer Transformers with logarithmic width can express the associative recall task. At the same time, the lower bound techniques for one-layer Transformers presented in Section D do not directly apply to the associative recall task and hence it is not straightforward to prove that Transformers must have two layers in order to perform the associative recall task.

The index lookup task serves a few purposes in our work. The result that it is in some sense easier for Transformers and difficult for recurrent models may not be very surprising based on the intuitive understanding of the architectures. Our results help theoretically formalize the intuitions that Transformers can use attention to arbitrarily retrieve tokens whereas recurrent models must compress the information in their hidden states. Secondly, the task helps us introduce the techniques that will be used for the constructions and lower bounds to obtain more general results in the later sections. Lastly, it serves as a simple task that separates one-layer Transformer and recurrent models. As described above, it is not straightforward to show that the associative recall task can or cannot be expressed by one-layer Transformers with small width but with the index lookup task, we have that one-layer Transformers can represent them efficiently.

## D   Lower Bounds for 1-layer Transformers

As described in Section B.2, and in keeping with real-world implementations, the outputs of intermediate computations are rounded to $p$-bit precision. Further in keeping with real implementations, and to avoid overflow in exponentiation, softmax is implemented by first subtracting the maximum logit,

as

$$\text{softmax}(A)_{i,j} = \frac{\exp(A_{i,j} - \max_l A_{i,l})}{\sum_{k=1}^{M} \exp(A_{i,k} - \max_l A_{i,l})}.$$

This is a popular approach to implementing softmax [9]; it ensures that the exponentiated intermediate results are in $[0, 1]$, avoiding possible overflow when exponentiating in finite precision.

**Theorem 4.** *Consider a one-layer Transformer $f \in \text{TF}^1_{m,p,H}$ operating over inputs of length $N$. Consider any disjoint subsets $S_A \cup S_B = \{1, \dots, N\}$, $S_A \cap S_B = \emptyset$. Assume Alice has access to $s_i$ for $i \in S_A$, and Bob has access to $s_i$ for $i \in S_B$. Then Alice and Bob can communicate $3m(p + \log N)H$ bits to compute the output $f(s_1 \dots s_N)$.*

We note that conceptually related arguments were used in Sanford et al. [51, Theorem 7] and Peng et al. [46, proof of Theorem 1]). Our approach here generalizes by stating this for *arbitrary* partitions over the input.

*Proof.* Without loss of generality, assume that $N \in S_A$, i.e., Alice has access to $x_N$.

In the *first step*, Alice sends $x_N$ to Bob using $dp$ bits of communication. Then, Alice and Bob compute for each head the attention logits for each position within their respective sets:

$$A_{N,i} := \langle Q(\mathbf{x}_N), K(\mathbf{x}_i) \rangle \tag{1}$$

These numbers are rounded to $p$ bits of precision.

In the *second step* of the protocol, Alice and Bob exchange $2p$ bits to determine $M := \max_i A_{N,i}$, and compute

$$\widehat{A}_{N,i} = A_{N,i} - M \tag{2}$$

for their respective positions $i \in S_A, S_B$. Note that, as $\widehat{A}_{N,i} \leq 0$, $\exp(\widehat{A}_{N,i}) \in (0, 1]$, so there are no overflow issues arising from the $p$-bit representation. Then they can individually compute

$$Z_A := \sum_{i \in S_A} \exp(\widehat{A}_{N,i}) \quad \text{(Alice)}$$

$$Z_B := \sum_{i \in S_B} \exp(\widehat{A}_{N,i}) \quad \text{(Bob)}$$

each with $p$ bits of precision; both numbers are in $[0, N]$, and at least one of them is $\geq 1$. As all intermediate computations are rounded to $p$ bits, $\exp(\widehat{A}_{N,i})$ is in $[0, 1]$ and rounded to $p$ bits, $Z_A, Z_B$ can be represented with $p + \log N$ bits.

In the *third step* of the protocol, they exchange $2(p + \log N)$ bits to exchange these. They then both have access to

$$Z := \sum_{j=1}^{N} \exp(\widehat{A}_{N,j}) = Z_A + Z_B \tag{3}$$

Here, we note that the definition of finite precision arithmetic in Appendix B.2 makes addition of nonnegative numbers associative; hence, the outcome here is independent of the partition $S_A, S_B$ and agrees with the result when directly summing the exponentiated logits.[4]

The result $Z$ is in $[1, N]$ as $\max_i \exp(\widehat{A}_{N,i}) = 1$. This permits Alice and Bob to compute the attention scores $\widehat{a}_i$:

$$\widehat{a}_i := \text{softmax}(A)_{N,i} = \frac{\exp(\widehat{A}_{Ni})}{\sum_{j=1}^{N} \exp(\widehat{A}_{Nj})} \tag{4}$$

each with $p$ bits of precision.

---

[4]A similar protocol still works in other bounded-precision schemes where addition is not associative; as all exponentiated logits are in $[0, 1]$, one can use extended precision with $p + \log N$ bits to perform the summation exactly and then round back to $p$ bits.

Then they both each compute:

$$U_A := \sum_{i \in S_A} \text{softmax}(A)_{N,i} V(\mathbf{x}_i) \quad \text{(Alice)}$$

$$U_B := \sum_{i \in S_B} \text{softmax}(A)_{N,i} V(\mathbf{x}_i) \quad \text{(Bob)}$$

with $p$ bits of precision. As each entry $\text{Att}(A)_{N,i}$, $\mathbf{x}_i$ has $p$ bits of precision, and $\sum_i \text{Att}(A)_{N,i} = 1$, the sums can be exactly represented at $2p$ bits of precision.

In the *fourth step* of the protocol, they exchange these, which amounts to $4mp$ bits of communication. Then they compute

$$\sum_{i=1}^{N} \text{softmax}(A)_{N,i} V(x_i) = U_A + U_B \tag{5}$$

and round it to $p$ bits of precision. From this, they can obtain the result.

In total, the four steps took $\leq 3m(p + \log N)$ bits of communication. Performing this protocol separately for every head leads to $\leq 3m(p + \log N)H$ bits of communication. $\qquad \square$

# E    Dyck with Bounded Depths

It is generally agreed that processing and comprehending natural language requires processing hierarchical structures [e.g. 13, 18]. The fundamental computational problem here consists of matching and relating material that matches hierarchically, even if it appears at a great linear distance. This problem is formalized by the family of Dyck languages [14]: languages of well-matched words over one or more types of parentheses. These languages formalize problems that can be solved with access to a stack, where opening parentheses are pushed and closing parentheses are popped. Beyond a fundamental model of hierarchical structure, they are of central importance in formal language theory, as any context-free language can be expressed in terms of Dyck languages [14]. Perhaps due to the boundedness of human memory [41], natural language tends to have more bounded levels of embedding [30, 10]. This has motivated the study of bounded-depth Dyck languages as plausible simple models of the hierarchical structure underlying language, with substantial interest in the abilities of neural architectures to model them [25, 65, 6].

We will primarily focus on Dyck-2 languages with depth at most $k$, denoted as Dyck-(2, k). The Dyck-2 language contains two types of brackets, such as round brackets '(', ')' and square brackets '[', ']'. The Dyck-(2, 2) language with depth at most 2 contains well-balanced parenthesis where for any prefix, the number of unbalanced parentheses can be at most 2. For instance, the string '( [ ] ) [ ]' has a depth at most 2 whereas the string '( [ [ ] ] )' has a depth of 3. We say a model recognizes a language if it can correctly classify whether or not a string belongs to the language. We will primarily focus on strings with maximum length $N$ and study how the size of a model depends on that.

Our main result in this section is that any 1-layer Transformer that can recognize Dyck-2 with bounded depths must have a width that grows linearly with the input length $N$. To show that, we will first show that the communication complexity of the Dyck-2 language with depth at most 2 is at least $N - 1$. The lower bound on the width of 1-layer Transformers will follow from the lower bound on the communication complexity of Dyck-(2, 2).

**Problem.** Let $\Sigma = \{\text{'(', ')'}, \text{'[', ']'}\}$ be the vocabulary of a language Dyck-(2, k). Let $\Sigma^n$ denote the set of all strings of length exactly $n$ and $\Sigma^{\leq n}$ denote the set of all strings of lengths up to $n$. The communication problem between Alice and Bob is defined as follows. Assume the length $N$ is even for this problem. For a string $x \in \Sigma^N$, Alice has the symbols in the odd indices of the string and Bob has the symbols in the even indices of the string. They have to compute the function $f_{\text{Dyck}} : \Sigma^{N/2} \times \Sigma^{N/2} \to \{0, 1\}$ which outputs 1 if the string $x$ is in the language Dyck-(2, 2) and outputs 0 otherwise.

For any function, the fooling set is defined in the following way,

**Definition 1.** [Fooling set] A fooling set for any function $f : \Sigma^{N/2} \times \Sigma^{N/2} \to \{0, 1\}$ is a set $S \subseteq \Sigma^{N/2} \times \Sigma^{N/2}$ and a value $b \in \{0, 1\}$ such that,

- For every $(x, y) \in S$, $f(x, y) = b$.

- For every two distinct pairs $(x_1, y_1), (x_2, y_2) \in S$, either $f(x_1, y_2) \neq b$ or $f(x_2, y_1) \neq b$.

The following is a well-known fact in communication complexity.

**Fact 4.** *For any function $f$, if there exists a fooling set of size $|S|$, then the communication complexity $C(f) \geq \log_2 |S|$.*

**Lemma 1.** *Suppose Alice and Bob have the symbols in the odd and even indices of a string $s \in \Sigma^N$ respectively. To each compute whether $s \in$ Dyck-(2, 2), they must exchange at least $N - 1$ bits.*

*Proof.* The proof follows from the fact that there exists a fooling set of size $2^{N-1}$ for the language Dyck-(2, 2) with strings of length $N$. The fooling set $S$ is constructed with all strings in $s = (x, y) \in \Sigma^N$ such that $f_{\text{Dyck}}(s) = 1$. Each string $s$ in $S$ satisfies:

$$s = (x, y) \quad \text{where} \quad x \in \Sigma_{\text{odd}}^{N/2}, y \in \Sigma_{\text{even}}^{N/2}, \text{ and } f_{\text{Dyck}}(x, y) = 1.$$

Here, $x = (x_1, x_2, \ldots, x_{N/2})$ and $y = (y_1, y_2, \ldots, y_{N/2})$ represent the sequences of symbols at odd and even indices, respectively.

**Constructing the fooling set.** Note that, if $x$ is the string of symbols in the odd indices of a string $s \in$ Dyck-(2, 2), then the string of symbols $y$ in the even indices such that $f_{\text{Dyck}}(x, y) = 1$ is unique. Suppose one is provided with the string $x = (x_1, \ldots, x_{N/2})$, then one can deterministically determine the symbols in the even indices $y = (y_1, \ldots, y_{N/2})$ in the following way. Iterate through the symbols in $x$, starting with $x_1$ which must be an open bracket. After the open bracket $x_1$, if there are one or more closing brackets $x_2, \ldots, x_K$ before encountering another open bracket $x_{K+1}$, then the symbols $y_1, \ldots, y_K$ can be constructed deterministically using the following mapping,

$$y_j = \begin{cases} \text{`('} & \text{if } x_{j+1} = \text{`)' for } j < K, \\ \text{`['} & \text{if } x_{j+1} = \text{`]' for } j < K, \\ \text{`)'} & \text{if } x_1 = \text{`(' for } j = K, \\ \text{`]'} & \text{if } x_1 = \text{`[' for } j = K. \end{cases}$$

In other words, the symbols $y_1, \ldots, y_{K-1}$ will be the open brackets corresponding to the closing brackets $x_2, \ldots, x_K$. After the symbol $x_1$, whenever you encounter another open bracket $x_{K+1}$ where $K > 0$, then the symbol $y_K$ will be the closing bracket corresponding to the symbol $x_1$. Once you have matched the first open bracket symbol $x_1$ with a closing bracket, you are bound to encounter another open bracket. Follow the same process until the end of the string and one can obtain the string $y$, such that $(x, y) \in$ Dyck-(2, 2).

Hence, if $x$ and $y$ are the symbols in the odd and even indices of a string $s \in$ Dyck-(2, 2), then placing any other string $y' \neq y$ in the even indices leads to a string which is not in Dyck-(2, 2). Our fooling set $S$ contains all strings of length $N$ in the language Dyck-(2, 2). Hence, by construction, we have that $f_{\text{Dyck}}(s) = f_{\text{Dyck}}(x, y) = 1$ for all $s = (x, y) \in S$ and

$$f_{\text{Dyck}}(x_1, y_2) = f_{\text{Dyck}}(x_2, y_1) = 0 \quad \text{for all} \quad (x_1, y_1) \neq (x_2, y_2) \in S.$$

Thus, such a set $S$ is a fooling set by Definition 1.

**Size of the fooling set.** One can show that the total number of strings of length $N$ in the language Dyck-(2, 2) is exactly $2^{N-1}$ with some elementary combinatorics. First, see that the number of Dyck-2 sequences of depth at most 1 and length $N$ is exactly $2^{N/2}$. This is because the string is made up of blocks of '()' and '[]', and hence there are two choices for every block. Then, consider a block of Dyck-2 string of length $k$ (where $k$ is even) which starts and ends with an open and closing bracket respectively. Moreover, the Dyck-2 string has a depth exactly 2, e.g. '[ () [] () ]'. Note that, the total number of such strings of length $k$ is exactly $2^{k/2}$ since there are two choices for the first bracket and there is a Dyck-2 string of depth 1 and length $k - 2$ inside it. For simplicity, we will call such strings as belonging to the language Dyckb-(2, 2) which is a subset of the language Dyck-(2, 2). In simple terms, strings of length $m$ in the subset Dyckb-(2, 2) have an overall depth 2 and have a depth 1 string of length $m - 2$ between the first symbol (open bracket) and the last symbol (closing bracket); for instance '[ () [] () () ]'.

The total number of Dyck-2 strings of depth at most 2 and length exactly $N$ can be computed as follows. Partition the indices into $k$ contiguous blocks where each partition is of an even length. Suppose the indices begin with 1, then the points of partition could be between 2 and 3, 4 and 5, and so on. For instance, a valid partition of the indices $[1, 2, \ldots, 8]$ into 3 blocks is $[1, 2]$, $[3, \ldots, 6]$, and $[7, 8]$. The total number of partition points is $N/2 - 1$. For any such partition, if the length of a block is 2 then that block can only have Dyck-2 strings of depth 1 and if the block is of length $\geq 4$, then consider all possibilities of Dyck-(2, 2) strings starting and ending with open and closing brackets respectively or in other words, strings in Dyckb-(2, 2) described earlier. If the number of partitions is $N/2 - 1$, then all possible strings have depth at most 1 and if the number of partitions is 0, then the entire string is in Dyckb-(2, 2).

See that, no matter how you partition the inputs, the number of possible strings at each block of length $m$ is $2^{m/2}$. Further if $P = \{p_1, \ldots, p_k\}$ denotes the set of lengths of each block in the partition, then the total number of strings with such a partition is $\prod_{i=1}^{k} 2^{p_i/2} = 2^{\sum_{i=1}^{k} p_i/2} = 2^{N/2}$. In other words, regardless of how the indices are partitioned if each block of size $> 2$ is required to be a string in Dyckb-(2, 2), then the total number of possible strings is $2^{N/2}$.

The total number of Dyck-2 strings of depth at most 2 can be computed by considering partitions of all sizes $k = 0, \ldots, N/2 - 1$ and all possible partitions for a given size. With this, we get the total number of valid strings in Dyck-(2, 2) of length $N$ to be

$$\sum_{k=0}^{\frac{N}{2}-1} \binom{\frac{N}{2}-1}{k} 2^{N/2} = 2^{N-1}$$

Hence from fact 4, it follows that the communication complexity of Dyck-(2, 2) is at least $N - 1$.

$\square$

While we have given a self-contained proof based on fooling sets, an alternative proof of Lemma 1 could proceed using varieties of finite monoids, by proving that the syntactic monoid of Dyck-(2, 2) is not in the variety DA, and then applying the result of Raymond et al. [49].

Using Lemma 1 and Theorem 4, it follows that any 1-layer Transformer that can recognize Dyck-(2, 2) must have a width that grows linearly with the input length.

**Theorem 10.** *Consider a one-layer transformer $f \in \mathrm{TF}_{d,p,H}^1$ deciding membership in Dyck-(2, 2). Then $dH \geq \frac{N-1}{3(p+\log N)}$.*

Suppose we allow the transition function in a recurrent model to be any arbitrary function as defined in Section 2. In that case, it naturally follows that such an RNN can represent any DFA with $k$ states with a hidden state of size $\log k$. Any bounded dyck language Dyck-(n, k) can be represented by a DFA with $O(2^k)$ states which is independent of the input length $N$. In particular, the language Dyck-(2, 2) considered here can be represented by a DFA with just 7 states. Hence, it follows that a recurrent model of constant size can represent the Dyck-(2, 2) language for arbitrary lengths in a finite precision setting. Additionally, prior works have described constructions of such recurrent models with practical activation functions such as Sigmoid [25] and ReLU [6]. Thus, the bounded Dyck language Dyck-(2, 2) provides us a separation between one-layer Transformer and recurrent models since constant-sized RNNs can represent them whereas one-layer Transformers must have at least linear width to represent them.

## F  Transformers and Boolean functions

**Communication protocols and lower bounds.** There are some notable differences between the types of communication complexity lower bounds we have for one-layer Transformers (Theorem 4) and for RNNs (Theorem 2). If an RNN can compute a function $f$ then Theorem 2 implies that there exists a one-way communication protocol with $mp$ bits over any *contiguous* partitions of the input. On the other hand, if a one-layer Transformer can compute a function $f$, then Theorem 4 implies the existence of a communication protocol with $O(mpH \log N)$ bits over *any partition* of the

input — but not one-way. Hence, the types of lower bounds that apply to these two architectures are different. For any function $f$ such as the INDEX problem, if the one-way communication complexity is lower bounded by $\Omega(N)$ over some contiguous partitions then it implies that for RNNs, the width $m = \Omega(N/p)$. However since the two-way communication complexity of such problems might be lower, those lower bounds need not apply to one-layer Transformers. For instance, the INDEX problem can be solved with $\log N$ bits of communication if two-way communication is allowed or in other words, Bob is allowed to send bits to Alice. Conversely, to prove lower bounds for Transformers computing a certain function $f$, proving a communication complexity lower bound for $f$ over any partition of inputs suffices. For a particular partitioning of inputs, we showed a lower bound on the communication complexity of bounded Dyck languages. While that result implies a lower bound on one-layer Transformers, the partitioning is not contiguous and hence does not apply to recurrent models. For Dyck-(2, 2), contiguous partitions are not a hard case, and in fact, communicating $\leq 2$ open brackets from the first half is sufficient. This is why the lower bound of Lemma 1 does not apply to RNNs. Despite these differences, for a large class of Boolean functions including functions such as Equality and Disjointness, the lower bounds apply to both of these architectures.

**Equality.** For convenience, we discuss the complement of the Equality function $\text{INEQ} = 1 - \text{EQ}(x)$ which is the inequality function. It does not influence any of our results or the constructions for Transformers but it helps make the intuitions for the construction clearer. The function $\text{INEQ} : \{0, 1\}^N \to \{0, 1\}$ is a Boolean function defined as,

$$\text{INEQ}(x) = \mathbb{I}[(x_1, \ldots, x_{N/2}) \neq (x_{N/2+1}, \ldots x_N)]$$

This can also be represented as a 2-DNF with $N/2$ terms,

$$\text{INEQ}(x) = (x_1 \wedge \neg x_{N/2+1}) \vee (\neg x_1 \wedge x_{N/2+1}) \vee \ldots \vee (x_{N/2} \wedge \neg x_N) \vee (\neg x_{N/2} \wedge x_N)$$

### F.1 Lower Bounds

We show that for any RNN that computes the INEQ function over $\{0, 1\}^N$ with a hidden state of size $m$ and $p$ bits of precision, it is necessary that $mp = \Omega(N)$. In other words, the size of the hidden state of the RNN grows linearly with the input length $N$. Similar to other results, this will be based on communication complexity but we also provide an alternate way to prove this lower bound based on the state complexity of DFAs that compute the INEQ function.

The result states that if any RNN with a hidden state of size $m$ over $p$-bit precision can simulate a DFA with $n$ states, then the representation size of the hidden state $mp$ must be at least $\log n$. There is one caveat related to this DFA-based result which does not apply to the communication complexity-based lower bounds. For the DFA-based result, while the transition function is allowed to be an arbitrary function, it has to be of the form $g(\mathbf{x}_t, \mathbf{h}_{t-1})$,i.e., the functions cannot be different based on the timestep $t$. It is unclear if the result still applies when the function is allowed to be different based on the timestep. However, since the transition function is allowed to be any arbitrary function, it still captures all the recurrent and state-space architectures used in practice. Additionally, this fairly simple technique could be used to prove lower bounds for RNNs based on lower bounds in formal language theory.

**Lemma 4.** *Let $\mathcal{A}$ be a DFA with $n$ states. Let $\mathcal{R}$ be an RNN such that for all $x \in \Sigma^*$, $\mathcal{A}(x) = \mathcal{R}(x)$. Then the size of the hidden state of the RNN: $mp \geq \log n$.*

*Proof.* Suppose an RNN $\mathcal{R}$ exists such that the dimension of its hidden state $m < \log n$. It is straightforward to see that we can construct another DFA $\mathcal{A}'$ from the RNN $\mathcal{R}$ such that the DFA $\mathcal{A}'$ accepts the same language as $\mathcal{A}$ but it has $2^{mp} < n$ states.

Create a state for each vector $\mathbf{h} \in \{0, 1\}^{mp}$ and assign the state corresponding to the vector $\mathbf{h}_0$ as the start state. For each state $q$ corresponding to vector $\mathbf{h}$, make $q$ a final state if $f(\mathbf{h}) = 1$. For each $\mathbf{h} \in \{0, 1\}^{mp}$ and each symbol $x \in \Sigma$, compute $\mathbf{h}' = g(\mathbf{x}, \mathbf{h})$. Add a transition between the state corresponding to $\mathbf{h}$ and $\mathbf{h}'$ using the symbol $x$. Hence, we get the entire DFA with the transition map and the set of start and final states that accept the same language.

Since we are given that the automata $\mathcal{A}$ is the minimum-sized DFA that accepts the language, the automata $\mathcal{A}'$ having fewer states and accepting the same language is a contradiction. Hence, such an RNN cannot exist and the dimension of the hidden state vector $m$ of the RNN must be at least $\log n$. $\square$

**Theorem 11.** *Any recurrent model with a hidden state of size $m$ over $p$-bits of precision that computes the* INEQ$(x)$ *for all $x \in \{0,1\}^N$ must have $mp \geq N/2$.*

*Proof.* We show two ways to prove the above statement. The first proof is based on the communication complexity of the Equality problem.

**Proof 1.** Suppose Alice and Bob have the input vectors $a, b \in \{0,1\}^{N/2}$ respectively and have access to an RNN that computes the INEQ function over $\{0,1\}^N$. Alice can first use the RNN starting with input $a_1$ and the vector $\mathbf{h}_0$ and iteratively update the hidden state until the input $a_{N/2}$. Alice can then send the vector $\mathbf{h}_{N/2}$ to Bob who can provide $b_1, \ldots, b_{N/2}$ as inputs and compute $y = \text{INEQ}(a \cdot b) = 1 - \text{EQ}(a, b)$. Since sending the hidden state vector $\mathbf{h}_{N/2}$ requires $mp$ bits, it must be that $mp \geq N/2$ due to Theorem 2.

**Proof 2.** The second proof uses the following relation between recurrent models and a DFA.

Let $\mathcal{A}_{INEQ}$ be the minimum-sized DFA that computes the INEQ function over $\{0,1\}^N$. When we say a DFA computes INEQ function over $\{0,1\}^N$, we mean that $\mathcal{A}_{INEQ}(x) = \text{INEQ}(x)$ for all $x \in \{0,1\}^N$ and $\mathcal{A}_{INEQ}(x)$ can be defined arbitrarily for $x \notin \{0,1\}^N$ to obtain the DFA with minimum number of states.

Note that, any DFA that agrees with INEQ over all $x$ in $\{0,1\}^N$ must have at least $2^{N/2}$ states. This follows from the fact that for any two distinct $x_1, x_2 \in \{0,1\}^{N/2}$ there is a distinguishing suffix $s \in \{0,1\}^{N/2}$ such that $\text{INEQ}(x_1 \cdot s) \neq \text{INEQ}(x_2 \cdot s)$. Hence, any DFA that agrees with INEQ on all inputs in $\{0,1\}^N$ must have at least $2^{N/2}$ states even if it is defined arbitrarily over other inputs $x \notin \{0,1\}^N$. From Lemma 4, it follows that $mp \geq N/2$.

$\square$

**Theorem 12.** *Any one-layer Transformer with a width $m$ and $H$ heads operating over $p$-bits of precision that computes the* INEQ$(x)$ *for all $x \in \{0,1\}^N$ must have $mpH = \Omega(N)$.*

The statement immediately follows from Theorem 4 and Fact 2.

### F.2 Transformer Construction for Equality

We now show how a log-sized 2-layer Transformer operating over log-precision numbers can compute the INEQ function over all Boolean inputs.

**Theorem 6.** *For any $N \in \mathbb{N}$, there exists a 2-layer Transformer $f \in \text{TF}^2_{m,p,2}$ where width $m = O(\log N)$ and precision $p = O(\log N)$ such that $f(\mathbf{x}) = \text{EQ}(\mathbf{x})$ for all $\mathbf{x} \in \{0,1\}^N$.*

We first describe the broad idea behind the construction. We will consider the input domain to be $\{-1,1\}^N$ and our goal is to compute INEQ$(x)$ for all $x \in \{-1,1\}^N$. This can also be formulated as,

$$\text{INEQ}(x) = (x_1 \oplus x_{\frac{N}{2}+1}) \vee (x_2 \oplus x_{\frac{N}{2}+2}) \vee \ldots \vee (x_{\frac{N}{2}} \oplus x_N)$$

Let $f^{(1)}_{\text{ineq}}$ denote the first layer of our construction $f_{\text{ineq}}$ that computes INEQ and let $f^{(1)}_{\text{ineq}}(x)_i$ denote the $i$th output vector of the first layer. Our construction will be such that for $i > N/2$, on the $i$th input the attention mechanism at the first layer will retrieve the $(i - N/2)$th input using tools described in Section B.3. With the MLP, the first layer $f^{(1)}_{\text{ineq}}(x)_i$ will compute $x_i \oplus x_{i-\frac{N}{2}}$ for each $i > N/2$ where $x_i \oplus x_{i-\frac{N}{2}} = 0$ if $x_i = x_{i-\frac{N}{2}}$ and is 1 otherwise. The second layer will then take an OR over all those values which will result in computing INEQ$(x)$.

*Proof.* Let $x = (x_1, \ldots, x_N)$ denote our input vector. The input to the Transformer model will include the positions as well $\tilde{x} = ((x_1, 1), \ldots, (x_N, N))$. Let $\mathbf{x}_i \in \mathbb{R}^d$ denote the embedding of the $i$th input $\tilde{x}_i$ where $d = 2 + 2k$. Here, $k = O(\log N)$ is the dimension of vectors $\mathcal{T}(1), \ldots, \mathcal{T}(N)$ described in Corollary 8.1. The input embeddings will contain four parts and will be of the following form

$$\mathbf{x}_i = \begin{cases} [x_i, 0, \mathcal{T}(i), \mathcal{T}(i), 1] & \text{if } i \leq N/2, \\ [x_i, 0, \mathcal{T}(i), \mathcal{T}(i - \frac{N}{2}), 1] & \text{otherwise.} \end{cases}$$

The query vectors $Q(\mathbf{x}_i) = \mathbf{x}_i W_Q \in$ will be the last part of the embeddings, i.e., $Q(\mathbf{x}_i) = [\mathcal{T}(i), 1]$ for $i \leq N/2$ and is $\mathcal{T}(i - N/2)$ for $i > N/2$. Similarly, the key vectors $K(\mathbf{x}_i) = [\mathcal{T}(i), -1/2]$ for all $i$. The value vector will be a $d$-dimensional vector $V(\mathbf{x}_i) = [0, x_i, \mathbf{0}_k, \mathbf{0}_k]$. The query, key, and value transformations can be implemented with block matrices containing zero and identity matrices similar to the one described in Section C.2.

By construction, the dot products $A_{i,j} = \langle Q(\mathbf{x}_i), K(\mathbf{x}_j) \rangle = \langle \mathcal{T}(i - N/2), \mathcal{T}(j) \rangle - 1/2$ will be such that for $i > N/2$,

$$A_{i,j} = \langle Q(\mathbf{x}_i), K(\mathbf{x}_j) \rangle \begin{cases} \geq 1/4 & \text{if } j = i - \frac{N}{2}, \\ \leq -1/4 & \text{otherwise.} \end{cases}$$

For $i < N/2$, the dot products $A_{i,j}$ will be greater than $1/4$ if $i = j$ and will be less than $-1/4$ for $i \neq j$. If this was a Transformer with the hard-attention mechanism, then, note that $\text{Att}(X)_i = [0, x_i, \mathbf{0}_k, \mathbf{0}_k]$ for $i \leq N/2$ and $\text{Att}(X)_i = [0, x_{i-N/2}, \mathbf{0}_k, \mathbf{0}_k]$ for $i > N/2$. With residual connections or another attention-head, the output of the attention block will include the original input as well which will lead to

$$\text{Att}(X)_i + \mathbf{x}_i = \begin{cases} [x_i, x_i, \mathcal{T}(i), \mathcal{T}(i), 1] & \text{for } i \leq N/2, \\ [x_i, x_{i-N/2}, \mathcal{T}(i), \mathcal{T}(i), 1] & \text{for } i > N/2. \end{cases}$$

Then a simple ReLU FFN can compute the XOR of the first two values of the output vector from the attention block. Hence, by construction, the output vector from the first layer $f_{\text{ineq}}^{(1)}(x)_i = [x_i \oplus x_{i-N/2}, \ldots]$ for $i > N/2$ and $[0, \ldots]$ for $i \leq N/2$.

**Second layer computing OR.** The second layer of the Transformer will compute the OR over the first coordinate of the input vector which is quite straightforward. Let the query vector $Q(\mathbf{x}_i^{(1)}) = 0$. The value vectors will be of the form $V(\mathbf{x}_i^{(1)}) = [(x_i \oplus x_{i-N/2}), 0, \ldots, 0]$ for $i > N/2$ and they will be zero vectors construction for $i \leq N/2$. Then, regardless of the keys, the dot products will be 0 for each position, and hence

$$\text{Att}(X)_N^{(2)} = \frac{1}{N} \sum_{i=N/2}^{N} (x_i \oplus x_{i-N/2}).$$

If $\text{Att}(X)_N^{(2)} \geq \frac{1}{N}$, then the FFN can output 1 and it can output 0 otherwise.

**Softmax attention.** We now describe how to retrieve the $x_{i-N/2}$ values with softmax attention in the first layer of the model. The approach is slightly different from the one used in Section C.2 and makes use of finite precision rounding.

Consider another construction that is identical to the construction described above with the exception that $\tilde{Q}(\mathbf{x}_i) = \eta Q(\mathbf{x}_i) = \eta [\mathcal{T}(i), 1]$. We show that for large enough $\eta$ and $i > N/2$, the weight $\text{softmax}(A)_{i,j}$ will be so close to 1 for $j = i - N/2$ that it will be rounded to 1. Similarly, it will be rounded to 0 for $j \neq i - N/2$. For $i \leq N/2$, the weight will be rounded to 1 for $i = j$ and will be rounded to 0 otherwise.

Recall that the finite precision implementation is parameterized by a large constant $K_c$ (cf. Appendix B.2). For $i > N/2$ and $j = i - N/2$, there exists an $\eta$ such that,

$$\left| 1 - \frac{\exp(\eta \langle Q(\mathbf{x}_i), K(\mathbf{x}_j) \rangle)}{\sum_{k=1}^{N} \exp(\eta \langle Q(\mathbf{x}_i), K(\mathbf{x}_k) \rangle)} \right| \leq \frac{1}{2N^{K_c}}.$$

For such an $\eta$, the weight $\text{softmax}(A)_{i,j}$ will be rounded to 1.

$$1 \geq \frac{\exp(\eta \langle Q(\mathbf{x}_i), K(\mathbf{x}_j) \rangle)}{\sum_{k=1}^{N} \exp(\eta \langle Q(\mathbf{x}_i), K(\mathbf{x}_k) \rangle)} \geq \frac{\exp(\frac{1}{4}\eta)}{\exp(\frac{1}{4}\eta) + (N-1)\exp(-\frac{1}{4}\eta)} \geq 1 - \frac{1}{2N^{K_c}}$$

$$\implies 2N^{K_c} \exp(\frac{1}{4}\eta) \geq (2N^{K_c} - 1)(\exp(\frac{1}{4}\eta) + (N-1)\exp(-\frac{1}{4}\eta))$$

$$\implies \exp(\frac{\eta}{4}) \geq (N-1)(2N^{K_c} - 1)\exp(-\frac{\eta}{4})$$

$$\implies \eta \geq 2\log\left((N-1)(2N^{K_c} - 1)\right)$$

Thus, for $\eta = \log N + K_c \log 2N$, the softmax attention weight at $j = i - N/2$ will be rounded to 1. Similarly, one may verify that if $\eta \geq K_c \log 2N$, then the weight $\mathrm{softmax}(A)_{i,j}$ for $j \neq i - N/2$ will be less than $1/2N^{K_c}$ and hence will be rounded to 0. Hence, for $\eta \geq \log N + K_c \log 2N$, the softmax attention will behave like hard attention, and as described earlier the Transformer will compute the INEQ function. This completes the proof.

$\square$

The scaling in the attention mechanism in the last part can also be implemented in almost exactly the same way as the construction for Theorem 1. Implementing it that way then requires the ReLU FFN to act as a threshold function. The scaling described above makes use of the finite precision setting to amplify the dot products to the point that it acts as hard attention.

### F.3 Representing more general class of Boolean functions

We now describe how the construction for Equality in Section F.2 can be extended to a class of Boolean functions namely, thresholds of at most $N$ $k$-SPARSE features. By threshold functions, we mean functions of the form $\mathsf{Th}_b : \{0,1\}^n \to \{0,1\}$ where for $x \in \{0,1\}^n$, the function is defined as $\mathsf{Th}_b(x) = \mathbb{I}[\sum_{i=1}^n x_i - b > 0]$. The function is parameterized by a constant $b$. For $b = n - 1$, the function effectively is the AND over all bits. Similarly, for $b = n/2$, the function outputs 1 if the majority of the input bits are 1 and outputs 0 otherwise.

A $k$-SPARSE function is simply a function whose output on any input $x \in \{0,1\}^n$ depends on at most $k$ indices of the input. More formally, a function $f : \{0,1\}^n \to \{0,1\}$ is $k$-SPARSE if there exist indices $1 \leq i_1 < i_2 < \ldots < i_k \leq n$ and a function $g : \{0,1\}^k \to \{0,1\}$, such that for every $x \in \{0,1\}^n$, $f(x_1, x_2, \ldots, x_n) = g(x_{i_1}, x_{i_2}, \ldots, x_{i_k})$. A $k$-SPARSE function can be any Boolean function (AND, OR, XOR, etc) with the constraint that it can depend on at most $k = O(1)$ bits of input.

We now define the class of threshold of at most $N$ $k$-SPARSE features denoted as $\mathrm{THRES}_{k,N}$. Let $g_I$ be a $k$-SPARSE function depending on $I \subset [N]$ indices where $|I| \leq k$. A function $f : \{0,1\}^N \to \{0,1\}$ is in the class $\mathrm{THRES}_{k,N}$ if it is of the form $f(x) = \mathsf{Th}_b(g_{I_1}(x), g_{I_2}(x), \ldots, g_{I_N}(x))$ where each $I_j \subset [N]$ and $|I_j| \leq k$ for all $j \in [N]$. Further $g_{I_j}(x) = 0$ for all $x \in \{0,1\}^N$ if the set $I_j = \emptyset$.

The following result states that for any function $h$ in the class of threshold of at most $N$ $k$-SPARSE features, there exists a two-layer Transformer with logarithmic width that can express the function $h$.

**Theorem 13.** *For any $N \in \mathbb{N}$ and any function $h \in \mathrm{THRES}_{k,N}$, there exists a 2-layer Transformer $f_{\mathrm{TF}} \in \mathrm{TF}^2_{m,p,k}$ with width $m = O(\log N)$, $H = k$ heads, and precision $p = O(\log N)$ such that $f_{\mathrm{TF}}(x) = h(x)$ for all $x \in \{0,1\}^N$.*

*Proof.* The result follows from a straightforward extension of the construction in Theorem 6. We will again remap the domain to $\{-1,1\}^N$. For this problem, we will also prepend the input $x \in \{-1,1\}^N$ with an additional beginning of sequence token [BOS]. Hence, the Transformer will receive $N + 1$ tokens $\mathbf{x}_0, \mathbf{x}_1, \ldots, \mathbf{x}_N$.

For any function $h \in \mathrm{THRES}_{k,N}$, the first layer of the Transformer will compute the $k$-SPARSE features, $g_{I_1}(x), g_{I_2}(x), \ldots, g_{I_N}(x)$. The second layer will then compute the threshold function $\mathsf{Th}_b$ over the $k$-SPARSE features.

**Computing Threshold.** The second layer is quite trivial to construct. Let $\mathbf{x}_1^{(1)}, \mathbf{x}_1^{(0)}, \ldots, \mathbf{x}_N^{(1)}$ be the output vectors of the first layer and inputs to the second layer. Suppose the $i$th output vector contains the $i$th $k$-SPARSE feature, $\mathbf{x}_i^{(1)} = [g_{I_1}(x), \ldots]$ for $i = 1, \ldots, N$ and $\mathbf{x}_0^{(1)} = \mathbf{0}$. If the query transformation is a null matrix, $Q(\mathbf{x}_N^{(1)}) = \mathbf{0}$, and the value transformation is such that $V(\mathbf{x}_i^{(1)}) = [g_{I_1}(x)]$, it follows that the output of the attention block will be $\frac{1}{N+1}\sum_{i=1}^N g_{I_i}(x)$. The

ReLU-based feedforward network can then be used to subtract with a constant $\frac{b}{N+1}$ and implement a threshold function to compute the desired output.

**Computing $k$-SPARSE features.** In the first layer, we compute the $k$-SPARSE features by making use of the almost orthogonal positional vectors $\mathcal{T}(0), \mathcal{T}(1), \ldots, \mathcal{T}(N)$ of size $r = O(\log N)$. From the construction for equality, it should be clear that using one attention head, one can attend to any desired position and retrieve a single bit. Extending that, using $k$ heads, we can retrieve $k$ bits from $k$ different positions. To compute $g_{I_i}(x)$, we will have input embeddings of size $1 + (k+1)r$. For the $i$th input $x_i$, the first coordinate will contain the input $x_i \in \{-1, 1\}$ and will contain 0 for the [BOS] token $x_0$. The next $r$ indices will contain the positional vector $\mathcal{T}(i)$ corresponding to the position $i$. The remaining $kr$ indices can be divided into $k$ blocks each of which will contain a positional vector. If the set $I_i$ contains indices $I_i^1, \ldots, I_i^k$, then the $k$ blocks will contain vectors $\mathcal{T}(I_i^1), \ldots, \mathcal{T}(I_i^k)$. If the set $I_i$ has less than $k$ indices, then the last $k - |I_i|$ blocks will have the positional vectors $\mathcal{T}(0)$.

See that if the input embeddings are designed as described above then one can obtain input bits $x_{I_i}$ in the output of the attention block at the $i$th token. The value vectors will be of $k$ dimension such that $V_h(\mathbf{x}_i) = x_i$ at the coordinate $h$ and is 0 everywhere else. The key transformation for each head can be $K(\mathbf{x}_i) = [\mathcal{T}(i)]$ containing the vector corresponding to the token's position. The query transformation is distinct for each head, and for the $j$th head, the query vector contains $j$th block from the last $k$ blocks containing the positional vectors for each index in the set $I_i$. If the query and key transformations are designed in such a way, then using the arguments described in Theorem 6, it can be seen that the output of the attention block at the $i$th position will be a $k$ dimensional vector containing $[x_{I_i^1}, \ldots, x_{I_i^k}]$. If the set $I_i$ has a size less than $k$, then the output vector will be followed by zeros after $|I_i|$ coordinates and will be a zero vector if $I_i = \emptyset$. Finally, the feedforward network can then compute the function $g_{I_i}(x)$ at every position which can be done by a network of constant size because $k = O(1)$ does not depend on $N$. The second layer can then compute the threshold over the $k$-SPARSE features as described earlier which leads to the desired output. $\qquad\square$

**Discussion.** The class of thresholds of $k$-SPARSE features contains certain functions of interest such as Disjointness and Equality as well as more general classes such as $k$-DNFs and $k$-CNFs with at most $N$ terms or clauses. As described earlier, the (In)Equality function can be also defined as $\mathrm{INEQ}(x) = (x_1 \oplus x_{\frac{N}{2}+1}) \vee \ldots \vee (x_{\frac{N}{2}} \oplus x_N)$. See that it contains $\frac{N}{2}$ 2-SPARSE features where the set of indices $I_i = \{i, i + \frac{N}{2}\}$ for $i = 1, \ldots, N/2$ and the feature function $g(a, b) = a \oplus b$ for $a, b \in \{0, 1\}$. Similarly, the Disjointness function can be represented as a 2-CNF. The complement of the Disjointness function can be described more simply as

$$(x_1 \wedge x_{\frac{N}{2}+1}) \vee (x_2 \wedge x_{\frac{N}{2}+2}) \vee \ldots \vee (x_{\frac{N}{2}} \wedge x_N)$$

which is a 2-DNF that outputs 0 if the first and second half of the input $x \in \{0, 1\}^N$ are disjoint and outputs 1 otherwise. Thus, it follows from Theorem 13 that two-layer Transformers with logarithmic width can represent functions such as Disjointness as well as $k$-DNFs and $k$-CNFs with at most $N$ terms or clauses.

### F.4 Difficulty of Deriving Communication-based Lower Bounds for 2-layer Transformers

Our lower bounds both for RNNs and for one-layer transformers are based on communication complexity arguments. Here, we provide evidence that other techniques may be needed to establish lower bounds for two-layer transformers by showing that, in a certain sense, no short communication protocol of the same kind as Theorem 4 can exist for Alice and Bob to obtain the output of any two-layer transformer. We start from the following lemma:

**Lemma 5.** *Assume $N$ is even. For every partition $S_A$, $S_B$ of $\{1, \ldots, N\}$ where $|S_A| = |S_B|$, there is a 2-layer transformer $f \in \mathrm{TF}_{d,p,2}^2$ with width $m = O(\log N)$ with precision $p = O(\log N)$ with $f(x) \in \{0, 1\}$ for each $x \in \{0, 1\}^N$, such that Alice and Bob, having access to $x_A$ and $x_B$ respectively, need to exchange $\geq \frac{N}{2}$ bits to compute $f(x)$.*

*Proof.* For any partitioning $S_A, S_B$, consider the task of determining whether $x_{S_A}$ and $x_{S_B}$ are identical. That is, define (for $x \in \{0, 1\}^N$):

$$f_{S_A, S_B}(x) = \begin{cases} 1 & \text{if } x_{S_A} \neq x_{S_B} \\ 0 & \text{else} \end{cases} \tag{6}$$

If Alice and Bob have access to $x_{S_A}$ and $x_{S_B}$, respectively, they need to exchange $\geq \frac{N}{2}$ bits to compute $f(x)$. Now by Theorem 6, there is a transformer $f_{ineq} \in \mathrm{TF}^2_{m,p,2}$ that computes $f_{[1,\ldots,n/2],[n/2+1,\ldots,n]}$, where $m = O(\log N)$ and $p = O(\log N)$. Now renumbering the positional encodings results in a transformer computing $f_{S_A,S_B}$.

$\square$

**Corollary 13.1.** *If there is any communication protocol by which Alice and Bob can compute the output for any two-layer Transformer, for any partition, with $O(mpHg(n))$ bits, then it must be the case that $g(n) = \Omega\left(\frac{N}{(\log N)^2}\right)$.*

*Proof.* Suppose there is a communication protocol such that for any two-layer Transformer and over any partition, the protocol can compute the output of the Transformer with

$$o\left(mpH\frac{N}{(\log N)^2}\right) \tag{7}$$

bits. But by Lemma 5, for each partition, there exists a two-layer Transformer with

$$mpH = O((\log N)^2) \tag{8}$$

such that Alice and Bob need to exchange $\geq N/2$ bits to compute its output. We obtain a contradiction.

$\square$

# G  Nearest Neighbors and Associative Recall

Recall from Section 5.2 that in the NSTNB task, a model is provided with a sequence of vectors and labels $(\mathbf{x}_1, y_1, \ldots, \mathbf{x}_{k-1}, y_{k-1}, \mathbf{x}_k)$ where $N/2 < k \leq N$ and the goal of the model is to predict the label corresponding to the nearest neighbor of $\mathbf{x}_k$ for each $k = \frac{N}{2} + 1, \ldots, N$. We first show that any recurrent model that performs this task must have a width or hidden state of size $\Omega(\frac{N}{p})$.

**Relation to Associative Recall.** The nearest neighbor task is closely related to the single and multi-query associative recall (MQAR) task introduced in Ba et al. [3] and Arora et al. [2]. In the associative recall task, a model receives a sequence $s_1, y_1, \ldots, s_{k-1}, y_{k-1}, s_k$ where the symbols $s_i$ belong to an alphabet $|\Sigma|$. Assume the symbols the $s_1, \ldots, s_{k-1}$ to be distinct and the labels $y_i \in \{0, 1\}$ for simplicity. The query input $s_k$ is a repetition of one of the preceding inputs. The goal of the model is to predict the label of the query input $s_k$ by finding the exact match from the context and producing the corresponding label. A model that can compute the nearest neighbor algorithm can perform these associative recall tasks by embedding the sequence of symbols $s_1, y_1, \ldots, s_{k-1}, y_{k-1}, s_k$ as a sequence of vectors $(\mathbf{x}_1, y_1, \ldots, \mathbf{x}_{k-1}, y_{k-1}, \mathbf{x}_k)$.

If a model receives a sequence $(\mathbf{x}_1, y_1, \ldots, \mathbf{x}_{k-1}, y_{k-1}, \mathbf{x}_k)$ where the vectors $\mathbf{x}_1, \mathbf{x}_2, \ldots, \mathbf{x}_{k-1}$ are distinct and the query vector $\mathbf{x}_k$ is a repetition then the task of applying nearest neighbor to predict the label for the query vector reduces to the single query associative recall task. Implementing the nearest neighbor algorithm is equivalent to finding the exact match for the query vector $\mathbf{x}_k$ and producing the label $y_k$ corresponding to that input. In the case, where the models are required to predict iteratively for all $\mathbf{x}_k$ for $k = \frac{N}{2} + 1, \ldots, N$, the task becomes equivalent to MQAR.

The MQAR task is a simplified version or a subset of the nearest neighbor task where the model is first provided with the prompt $(\mathbf{x}_1, y_1, \ldots, \mathbf{x}_{N/2}, y_{N/2})$ and the subsequent $N/2$ input vectors are a permutation of the first $N/2$ vectors. In other words, for $N/2 < k \leq N$ and a sequence $(\mathbf{x}_1, y_1, \ldots, \mathbf{x}_{k-1}, y_{k-1}, \mathbf{x}_k)$, the model has to find the exact match of $\mathbf{x}_k$ in the first $N/2$ vectors and output the corresponding label. It is straightforward to see that any model that can perform the nearest neighbor task can also perform the MQAR task. Assume the size of the alphabet $|\Sigma| = \Theta(N)$ The embedding of each symbol $s_i \in \Sigma$ can be a distinct vector from the hypercube $\{-\frac{1}{\sqrt{d}}, \frac{1}{\sqrt{d}}\}^d$ where $d = \lceil \log |\Sigma| \rceil = O(\log N)$. The embeddings can be thought of as the normalized binary encodings of each symbol. Thus, a model receives a sequence of vectors $(\mathbf{x}_1, y_1, \ldots, \mathbf{x}_{k-1}, y_{k-1}, \mathbf{x}_k)$ corresponding to a sequence of symbols $s_1, y_1, \ldots, s_{k-1}, y_{k-1}, s_k$. Since the query vectors $\mathbf{x}_k$ for $k > N/2$ are repetitions, there is an exact match for each of them in $\mathbf{x}_1, \ldots, \mathbf{x}_{N/2}$. Suppose the exact match for a query $\mathbf{x}_k = \mathbf{x}_{j^*}$, then their dot products $\langle \mathbf{x}_k, \mathbf{x}_{j^*} \rangle = 1$ and the dot product with

every other vector $\langle \mathbf{x}_k, \mathbf{x}_i \rangle \leq 1 - \frac{1}{\log N}$. Since the margin between the dot product with the nearest neighbor and other vectors satisfy $\frac{1}{\gamma} = O(\log N)$ and all the input embeddings have unit norms, the problem satisfies the assumptions of the nearest neighbor task.

## G.1 Lower Bounds for Recurrent Models

**Theorem 8.** *Any recurrent model with a hidden state of width $m$ with $p$-bits of precision that can perform the nearest neighbor task for all inputs of length $N$ must have $m \geq N/2p$.*

*Proof.* The proof is via a reduction from the disjointness problem. We show that the lower bound is true even for the restricted problem of multi-query associative recall (MQAR). Recall that in the MQAR task the sequence of query inputs $\mathbf{x}_{\frac{N}{2}+1}, \ldots, \mathbf{x}_N$ is a permutation of the sequence of labelled vectors $\mathbf{x}_1, \ldots, \mathbf{x}_{N/2}$.

If there exists a recurrent model $\mathcal{R}$ that can solve the MQAR task, then we show that Alice and Bob can use it to follow a communication protocol and compute the DISJ function.

The communication protocol is as follows. Alice and Bob have two Boolean vectors $a, b \in \{0, 1\}^{N/2}$ respectively. Both of them know the description of the recurrent model $\mathcal{R}$. Alice and Bob have decided on a set of $N/2$ vectors $\mathbf{v}_1, \ldots, \mathbf{v}_{N/2}$ in $\mathbb{S}^{d-1}$ that they will use in the communication protocol. Choosing any $N/2$ distinct vectors from the hypercube $\{-\frac{1}{\sqrt{d}}, \frac{1}{\sqrt{d}}\}^d$ will suffice and also satisfy the assumptions mentioned in Section 5.2.

To reiterate, both Alice and Bob have the model parameters/description and have decided on a set of $N/2$ unit vectors as a part of their communication protocol. Alice then uses the recurrent model $\mathcal{R}$ and provides the sequence of pairs $(\mathbf{v}_i, a_i)$ in any arbitrary order. Alice then sends the hidden state vector $\mathbf{h}_N$ to Bob. Bob then uses the model $\mathcal{R}$ and iteratively provides $\mathbf{v}_i$ vectors along with the generated output in any order. Bob knows that the output produced by the recurrent model for the vector $\mathbf{v}_i$ corresponds to the value $a_i$. Hence, Bob obtains the entire vector $a$ and computes $\text{DISJ}(a, b)$.

Since Alice sent the hidden state vector, which requires $mp$ bits, and they could compute $\text{DISJ}(a, b)$ over $\{0, 1\}^{N/2}$ by exchanging $mp$ bits, by Fact 1 we have that $mp \geq N/2$. $\qquad \square$

## G.2 Transformer Construction for Nearest Neighbor

We now show how a log-sized 2-layer Transformer operating over log-precision numbers can compute the NSTNB function over all inputs that satisfy the constraints described in Section 5.2. Recall the assumptions that

- Norm. All vectors $\mathbf{x}_i$ have unit norms.
- Margin. For any $N/2 < k \leq N$, and let $j^* = \arg\max_{i \in [k-1]} \mathbf{x}_k^T \mathbf{x}_i$, then $\mathbf{x}_k^T \mathbf{x}_{j^*} \geq \mathbf{x}_k^T \mathbf{x}_i + \gamma$ for any $i \neq j^*$.

**Theorem 7.** *For any $N \in \mathbb{N}$, there exists a 2-layer Transformer $f_{NN} \in \text{TF}_{m,p,2}^2$ with width $m = O(\log N)$ and precision $p = O(\log N)$ such that $f_{NN}$ computes the nearest-neighbor task all sequences of length at most $N$ satisfying the assumptions above.*

For $N/2 < k \leq N$, the model will be provided the sequence $(\mathbf{x}_1, y_1, \ldots, \mathbf{x}_{k-1}, y_{k-1}, \mathbf{x}_k)$ where $\mathbf{x}_i \in \mathbb{R}^{d'}$ vectors contain the input points and $y_i$s contain the labels. For clarity, we will refer to the embedding of all inputs as $\mathbf{z}_i \in \mathbb{R}^d$ where $\mathbf{z}_{2i-1} = \phi(\mathbf{x}_i, 2i-1)$ is the embedding of input points for $i = 1, \ldots, k$. Similarly, the vectors $\mathbf{z}_{2i} = \phi(y_i, 2i)$ will contain the embedding for the corresponding labels. Hence, technically the sequence of input vectors to the Transformer model will be $(\mathbf{z}_1, \mathbf{z}_2, \ldots, \mathbf{z}_{2k-1})$.

**Overview of Idea.** The explicit construction is a bit tedious but the key ideas are straightforward to understand. The construction itself does not rely on input sequences of fixed lengths, unlike the one for the Equality problem.

Recall that for an input sequence $(\mathbf{z}_1, \mathbf{z}_2, \ldots, \mathbf{z}_{2k-1})$, the odd indices $(2i-1)$ correspond to the inputs $\mathbf{x}_i$ and the even indices $(2i)$ contain the label information $y_i$. The final input $\mathbf{z}_{2k-1}$ contains

the query input $\mathbf{x}_k$ and the goal of the model is to find the nearest neighbor input $\mathbf{x}_{j^*}$ and output the label corresponding to that $y_{j^*}$.

Intuitively, a two-layer Transformer can do that in the following way: the first layer can find the nearest neighbor $\mathbf{x}_{j^*}$ and retrieve the position of the label $y_{j^*}$. The second layer can then attend over that position and produce the desired label.

**Challenges.** There are a few challenges to executing this strategy which our construction will address. If the input embeddings were identical to the input vectors, then $\mathbf{x}_k$ would have maximum dot product with itself and not with its nearest neighbor $\mathbf{x}_{j^*}$. Second, if you remove that, even then the dot product with some label vector could be larger than the dot product with the nearest neighbor (which could be close to $-1$). Lastly, suppose by design you have the maximum dot product with the desired input, you still need to retrieve the required information in a useful way since we are working with softmax attention which will also put weight over other inputs.

**Key ideas.** One can think of the vectors $\mathcal{T}(1), \ldots, \mathcal{T}(N)$ as some form of positional or address vectors of dimension $O(\log N)$. All vectors $\mathbf{z}_{2i-1}$ corresponding to inputs $\mathbf{x}_i$ will have the address vectors of their next position $\mathcal{T}(2i)$. Along with that, all the vectors $\mathbf{z}_i$ will also have their own address/position vectors $\mathcal{T}(i)$. If in the first layer, the model can attend over the vector $\mathbf{z}_{2j^*-1}$ with a high attention weight, then it will be able to retrieve the required address vector $\mathcal{T}(2j^*)$ which it can then use in the second layer to retrieve the required label.

The input embeddings are designed in such a way that the maximum dot product will be with the desired input vector $\mathbf{x}_{j^*}$ or more specifically $\mathbf{z}_{2j^*-1}$. The embeddings are such that the dot products between the query vector of input $\mathbf{z}_{2k-1}$ and the key vectors of all other input vectors $\mathbf{z}_i$ will be of the following form,

$$\langle Q(\mathbf{z}_{2k-1}), K(\mathbf{z}_{2i-1}) \rangle = \langle \mathbf{x}_k, \mathbf{x}_i \rangle - c_1 \langle \mathcal{T}(2k-1), \mathcal{T}(2i-1) \rangle \quad \text{for } i = 1, \ldots, k.$$

See that for $i \neq k$, $\langle Q(\mathbf{z}_{2k-1}), K(\mathbf{z}_{2i-1}) \rangle \approx \langle \mathbf{x}_k, \mathbf{x}_i \rangle$ whereas for $i = k$, $\langle Q(\mathbf{z}_{2k-1}), K(\mathbf{z}_{2i-1}) \rangle < -2$ or much smaller based on the constant $c_1$. Hence, attention weight will be smaller on the query input itself compared to other inputs.

Secondly, for $i = 1, \ldots, k-1$, the dot products with the label vectors will be of the following form,

$$\langle Q(\mathbf{z}_{2k-1}), K(\mathbf{z}_{2i}) \rangle = \langle \mathbf{x}_k, \mathbf{0} \rangle - c_1 \langle \mathcal{T}(2k-1), \mathcal{T}(2i-1) \rangle - c_2 \approx -c_2 \quad \text{for } i = 1, \ldots, k.$$

which will be small depending on the constant $c_2$. Hence, the dot product will be maximum with input with the nearest neighbor $\mathbf{x}_{j^*}$ with a margin $\Omega(\gamma)$.

To retrieve and use $\mathcal{T}(2j^*)$ in the next layer, we do not need to hard-attend on $\mathbf{z}_{2j^*-1}$. Since the address vectors are of the form $\mathcal{T}(i) \in \{-\frac{1}{\sqrt{k}}, \frac{1}{\sqrt{k}}\}^k$, we only retrieve the corresponding sign vectors $\mathcal{T}_s(i) = \sqrt{k}\mathcal{T}(i) \in \{-1, 1\}^k$. Since the attention weight will be maximum on $\mathbf{z}_{2j^*-1}$ with a margin, we can scale the query vectors to increase the weight to a sufficient value and then use ReLU as a form of threshold to obtain $\mathcal{T}_s(2j^*)$. The second layer is then straightforward and will retrieve the desired label.

*Proof.* We will now describe the explicit construction for Transformers to compute nearest neighbors.

**Input Embeddings.** The embedding vectors are defined in the following way,

$$\phi(\mathbf{x}_i, 2i-1) = \mathbf{z}_{2i-1} = [\mathbf{x}_i, 0, 1, \mathcal{T}(2i-1), \mathcal{T}(2i), \mathbf{0}_k] \tag{9}$$

$$\phi(y_i, 2i) = \mathbf{z}_{2i} = [\mathbf{0}_{d'}, y_i, -2, \mathcal{T}(2i), \mathcal{T}(2i), \mathbf{0}_k].$$

Here, the vectors $\mathcal{T}(i) \in \{-\frac{1}{\sqrt{k}}, \frac{1}{\sqrt{k}}\}^k$ are JL transformations as described in Section B.3 and hence $k = O(\log N)$. The dimension of the embedding vectors $d = 3k + d' + 2 = O(\log N)$. The vectors $\mathcal{T}(1), \ldots, \mathcal{T}(2N)$ are such that,

$$\langle \mathcal{T}(i), \mathcal{T}(j) \rangle = \begin{cases} 1 \pm \gamma/100 & \text{if } i = j, \\ 0 \pm \gamma/100 & \text{otherwise.} \end{cases}$$

**Query, key and value vectors.** The query and key transformations in the first layers are designed in the following way,

$$Q(\mathbf{z}_{2k-1}) = [\mathbf{x}_k, 1, -10\mathcal{T}(2k-1)]$$
$$K(\mathbf{z}_{2i-1}) = [\mathbf{x}_i, 3, \mathcal{T}(2i-1)] \quad \text{for } i = 1, \ldots, k,$$
$$K(\mathbf{z}_{2i}) = [\mathbf{0}_d, -2.3, \mathcal{T}(2i)] \quad \text{for } i = 1, \ldots, k.$$

The value vectors will only have the 5th part ($\mathcal{T}(2i)$) in the last slot and all other values will be 0. Let $\mathcal{T}_s(i) = \sqrt{k}\mathcal{T}(i) \in \{-1, 1\}^k$, that is, $\mathcal{T}_s(i)$ contains the signs of the vector $\mathcal{T}(i)$. The value vector will be of the form $V(\mathbf{z}_{2i-1}) = [\mathbf{0}_{d'}, 0, 0, \mathbf{0}_k, \mathbf{0}_k, 2\mathcal{T}_s(2i)]$ and $V(\mathbf{z}_{2i}) = [\mathbf{0}_{d'}, 0, 0, \mathbf{0}_k, \mathbf{0}_k, 2\mathcal{T}_s(2i)]$ for all $i = 1, \ldots, N$. The goal of the attention in the first layer is to retrieve the vector $\mathcal{T}_s(2j^*)$ corresponding to the label of the nearest neighbour.

**Inner products.** The design of such query and key transformation ensures that for an input query $\mathbf{x}_k$, the dot product is maximum with its nearest neighbour $\mathbf{x}_{j^*}$ and not with itself or any of the embeddings of the labels $y_i$s.

The inner products $A_{2k-1, 2i-1} = \langle Q(\mathbf{z}_{2k-1}), K(\mathbf{z}_{2i-1}) \rangle$ have the following form,

$$\langle Q(\mathbf{z}_{2k-1}), K(\mathbf{z}_{2i-1}) \rangle = \langle \mathbf{x}_k, \mathbf{x}_i \rangle + 2 - 10\langle \mathcal{T}(2k-1), \mathcal{T}(2i-1) \rangle$$

$$= \begin{cases} \langle \mathbf{x}_k, \mathbf{x}_i \rangle + 3 \pm \gamma/10 & \text{if } i \neq k, \\ \langle \mathbf{x}_k, \mathbf{x}_i \rangle + 3 - 10 \pm \gamma/10 & \text{if } i = k. \end{cases}$$

$$\implies \langle Q(\mathbf{z}_{2k-1}), K(\mathbf{z}_{2i-1}) \rangle \begin{cases} \geq 1 & \text{if } i \neq k, \\ \leq -5 & \text{if } i = k. \end{cases}$$

Similarly, the inner product with the label vectors $\mathbf{z}_{2i}$s is less than 0 for all $i$ as well,

$$\langle Q(\mathbf{z}_{2k-1}), K(\mathbf{z}_{2i}) \rangle = 0 - 6 \pm \gamma/10 \leq -5$$

To summarize, the query and key vectors are such that for the query input $\mathbf{z}_2k - 1$, the dot product with itself $\langle Q(\mathbf{z}_{2k-1}), K(\mathbf{z}_{2k-1}) \rangle \leq -5$ and the dot product with all label vectors containing $y_i$ is $\langle Q(\mathbf{z}_{2k-1}), K(\mathbf{z}_{2i}) \rangle \leq -5$. The remaining dot products are with the vectors of interest which include the nearest neighbour input point,

$$\langle Q(\mathbf{z}_{2k-1}), K(\mathbf{z}_{2i-1}) \rangle = \langle \mathbf{x}_k, \mathbf{x}_i \rangle \pm \gamma/10 \quad \text{for } i = 1, \ldots, k-1.$$

Suppose $j^* = \arg\max_{i \in [k-1]} \mathbf{x}_k^T \mathbf{x}_i$ is the index for the input point which has the minimum L2 distance or maximum inner product with the query vector $\mathbf{x}_k$. Then we have that,

$$\langle Q(\mathbf{z}_{2k-1}), K(\mathbf{z}_{2j^*-1}) \rangle - \langle Q(\mathbf{z}_{2k-1}), K(\mathbf{z}_{2i-1}) \rangle \geq \gamma - \gamma/5 \qquad (10)$$

for all $i \neq j^*$. This indicates the maximum inner product will be with its nearest neighbor and all other inner products will have a margin of at least $\tau = \frac{4}{5}\gamma$.

If we have a Transformer with a hard-attention mechanism, then it will attend only to the input which is the nearest neighbor of the query vector $\mathbf{x}_k$. However, with softmax attention, it is non-trivial to only attend over a single input.

The embedding of all input vectors $\mathbf{x}_i$ contains a vector $\mathcal{T}(2i)$ (see Eq. 9) which serves as a key vector to retrieve the label following that input vector. Note that, we only need to retrieve the signs of the vector $\mathcal{T}(2i)$ since the vectors are of the form $\{-\frac{1}{\sqrt{k}}, \frac{1}{\sqrt{k}}\}^k$. We can then use it to retrieve the label of the corresponding input in the next layer.

**Retrieving with softmax.** We show using softmax attention and a ReLU FFN we can retrieve the required $\mathcal{T}(2j^*)$. In particular, we will obtain $\mathcal{T}_s(2j^*) = \sqrt{k}\mathcal{T}(2j^*) \in \{-1, 1\}^k$. As shown earlier, if $j^* = \arg\max_{i \in [k-1]} \mathbf{x}_k^T \mathbf{x}_i$, then the maximum dot product $\langle Q(\mathbf{z}_{2k-1}), K(\mathbf{z}_{2j^*-1}) \rangle \geq \langle Q(\mathbf{z}_{2k-1}), K(\mathbf{z}_{2i-1}) \rangle + \tau$ is greater by a margin $\tau = \frac{4}{5}\gamma$. The value vector corresponding to $\mathbf{z}_{j^*}$, i.e., $V(\mathbf{z}_{j^*}) = [0, \ldots, 0, 2\mathcal{T}_s(2j^*)]$ has the key vector which will allow the next layer to retrieve the required label.

The basic idea is that even if at least $9/10$ of the attention weight is on $V(\mathbf{z}_{j^*})$ and essentially $2\mathcal{T}_s(2j^*)$ then that suffices to preserve the signs using a ReLU FFN.

Let $\sigma(a)$ be a function such that $\sigma(a) = 1$ for $a > 1$, $\sigma(a) = -1$ for $a < -1$, and $\sigma(a) = a$ for $-1 \leq a \leq 1$. See that $\sigma(a)$ can easily be implemented by a ReLU FFN since it is essentially $\text{ReLU}(x+1) - \text{ReLU}(x-1) - 1$.

Without loss of generality, let's say $2\mathcal{T}_s(2j^*)_1 = +2$. If the attention weight on it is $9/10$ and the remaining $1/10$ weight is distributed among the rest of the inputs, then in the worst case that value will be $\frac{9}{10}2 - \frac{1}{10}2 \geq 1$. Hence, applying $\sigma(\cdot)$ over the value will result in $+1$. The same goes for the case when $2\mathcal{T}_s(2j^*)_1 = -2$ and for other indices of $2\mathcal{T}_s(2j^*)$. Thus, it suffices to show that the attention weight over $V(\mathbf{z}_{2j^*-1})$ can be greater than $9/10$ since it implies that with the ReLU FFN, the output for $\mathbf{z}_{2k-1}$ in the first layer will contain $\mathcal{T}_s(2j^*)$.

Consider another construction which is identical to the construction described above with the exception that $\tilde{Q}(\mathbf{z}_i) = \eta Q(\mathbf{z}_i)$. We show that for large enough $\eta$, the weight $\text{softmax}(A)_{2k-1,2j^*-1}$ will be greater than $9/10$ and the remaining weights combined will be less than $1/10$. Let $\beta = \langle \tilde{Q}(\mathbf{z}_{2k-1}), K(\mathbf{z}_{2j^*-1}) \rangle$ and $Z \in \mathbb{R}^{N \times d}$ contain vectors $(\mathbf{z}_1, \ldots, \mathbf{z}_{2k-1})$. Then, for any $r \in \mathbb{N}$,

$$\text{softmax}(\tilde{Q}(\mathbf{z}_{2k-1})K(Z)^T)_{2j^*-1}$$

$$= \frac{\exp(\eta \langle Q(\mathbf{z}_{2k-1}), K(\mathbf{z}_{2j^*-1}) \rangle)}{\exp(\eta \langle Q(\mathbf{z}_{2k-1}), K(\mathbf{z}_{2j^*-1}) \rangle) + \sum_{p \neq 2j^*-1} \exp(\eta \langle Q(\mathbf{z}_{2k-1}), K(\mathbf{z}_p) \rangle)}$$

$$\geq \frac{\exp(\eta\beta)}{\exp(\eta\beta) + (2k-1)\exp(\eta(\beta-\tau))} \geq \frac{\exp(\eta\beta)}{\exp(\eta\beta) + 2N\exp(\eta(\beta-\tau))} \geq \frac{r-1}{r}$$

$$\implies \exp(\eta\beta) \geq (r-1)(2N)\exp(\eta(\beta-\tau))$$

$$\implies \eta \geq \frac{1}{\tau}\log(r-1)2N.$$

Hence, for $r = 10$, if $\eta = \frac{5}{4\gamma}\log 18N$ then the attention weight on the $2j^*-1$th input vector will be greater than $9/10$. Similarly, one can verify that for $\eta = \frac{5}{4\gamma}\log 18N$, the attention weight for the rest of the inputs combined is at most $1/10$.

**Output of the first layer.** Let $\mathbf{z}_i^{(1)}$ denote the output of the first layer on the $i$th input vector. By construction, with MLP and residual connection after the attention block, the output of the first layer is such that,

$$\mathbf{z}_{2k-1}^{(1)} = [\mathbf{x}_k, 0, 1, \mathcal{T}(2k-1), \mathcal{T}(2k), \mathcal{T}_s(2j^*)]$$

$$\mathbf{z}_{2i-1}^{(1)} = [\mathbf{x}_i, 0, 1, \mathcal{T}(2i-1), \mathcal{T}(2i), \ldots] \quad \text{for } i = 1, \ldots, k-1$$

$$\mathbf{z}_{2i}^{(1)} = [\mathbf{0}_{d'}, y_i, -2, \mathcal{T}(2i), \mathcal{T}(2i), \ldots] \quad \text{for } i = 1, \ldots, k-1.$$

**Second Layer.** Since we have the address/key vector $\mathcal{T}_s(2j^*)$ for the target label as the input in the first layer, it is straightforward to apply attention and $\sigma(\cdot)$ with ReLU FFN to produce the desired label.

See that if the query vector $Q(\mathbf{z}_{2k-1}^{(1)}) = [\frac{1}{k}\mathcal{T}_s(2j^*)]$, and the key vectors contain the 4th part of the input vector, that is, $K(\mathbf{z}_i^{(1)}) = [\mathcal{T}(i)]$, then the dot product will be greater than $1 - \gamma/100$ with the desired input at $2j^*$ and will be less than $\gamma/100$ for the rest of the inputs. The value vectors will be assigned the second part of the input $V(\mathbf{z}_{2i}^{(1)}) = [\mathbf{0}_{d'}, y_i, 0, \ldots, 0]$ and will $V(\mathbf{z}_{2i-1}^{(1)}) = [0, \ldots, 0]$ for all $i = 1, \ldots, k$. Using the techniques used for the first layer, it is straightforward to see that a scaling of the query vector and applying $\sigma(\cdot)$ to the output will produce the desired label $y_{j^*}$.

$\square$

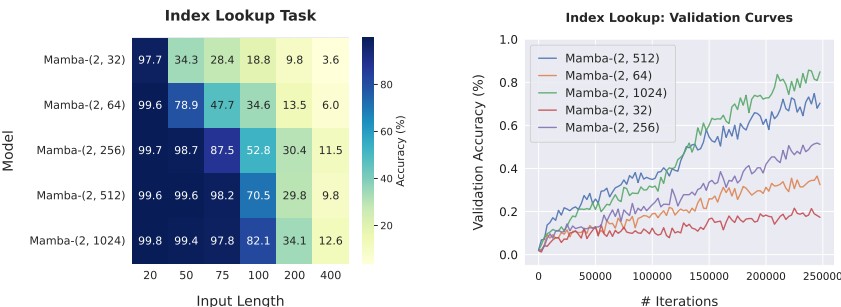

Figure 3: Performance of Mamba on the Index Lookup task across various lengths and widths. See Section H.1 for more details.

# H  Empirical Analysis: Additional Details and Experiments

In this section, we discuss the details of implementation and data generation for experiments described in Section 6. Further, we discuss some additional experiments on the string equality task.

**Implementation and hyperparameters.** All of our implementations for experiments are based on PyTorch [44]. For our experiments with Transformers, we use the Huggingface Transformers library [63] with the GPT-2 backbone as well as our own custom implementation. We use PyTorch's standard or in-built implementation for experiments with LSTMs. For state-space models like Mamba [21] and Diagonal state-space models [22], we use the official implementation provided by the authors for our experiments. For RetNet [56] and Linear Transformers, we use our own custom implementation.

For each task, we tune the models across several hyperparameters and report the results based on the best-performing model. We use grid search to tune the models for each task. For each architecture excluding LSTMs, we tuned across depths $\in \{1, 2, 4, 6\}$ and widths $\{64, 128, 256, 512\}$. Since LSTMs with large depths are hard to train [43], we only consider LSTMs with depths up to 3. For Transformers and linear Transformer variants, we tune the heads across $\{4, 8\}$. For all models, we tune the learning rate across $\{0.01, 0.005, 0.001, 0.0005, 0.0001, 0.00005, 0.000001\}$. We primarily use Transformers with absolute encodings for the result presented in the main paper. We train the models for up to 250k steps unless they achieve (almost) perfect validation accuracy earlier. After training, we evaluate the models on a fresh set of 5000 examples.

Note, however, that the focus of our paper is slightly different and for some problems, we are interested in the behavior of small-sized (width or depth) models of one architecture and a relatively larger model for another architecture. For instance, in the index lookup task, we are concerned with how well one-layer Transformers with small widths fare against relatively larger recurrent models. Additionally, our focus is on the comparison of different architectures of fixed sizes across lengths and not primarily on how well models of the scale used in practice perform these tasks on much larger lengths. Hence, we do not consider models with much larger depth or width.

**Compute.** All our experiments were conducted using 8 NVIDIA Tesla V100 GPUs each with 16GB memory and 16 NVIDIA GTX 1080 Ti GPUs each with 12GB memory. Each run for 500k steps could take between 1 hour and 16 hours depending on the length of the inputs and the size of the models. Some runs are much shorter if the model achieves high accuracy quite early (e.g. on lengths 20). While each individual run does not take a significant amount of time, tuning the models, particularly for the tasks where they fail to learn requires several runs and consequently a much longer duration. We estimate that all the runs across hyperparameters, architectures, and input lengths took $\leq 1200$ GPU hours in total on the GPUs mentioned above.

## H.1  Additional Experiments and Data Generation

**Index Lookup Task.** The input sequences are comprised of symbols in $\Sigma$ and a positional token in $[N]$. In our experiments, the size of the set of symbols is $|\Sigma| = 64$ and $N$ varies from 20 to 400. To create each example, we first sample a length $k$ uniformly between 10 and $N$ (20 - 400) and then sample $N$ symbols independently from $\Sigma$ uniformly at random. Lastly, we sample a position between

1 and $k$ uniformly at random and append it to the sequence to produce a labeled example for training or evaluation. During evaluation, the model is tested on sequences of length exactly $N$.

**Bounded Dycks.** For the Dyck-(2, 2) task, we generate the examples in such a way that with 0.5 probability, the generated string is well-balanced with depth at most 2, and with 0.5 probability it does not belong to the language and has label 0. The generated strings are of length exactly $N$ in both cases. To generate positive examples we use the following strategy: we iterate over $N - 2$ steps and for each step, we check whether the current depth of the stack is $< 2$. If the depth of the stack is 2, the sequence continues with the closing bracket corresponding to the open bracket in the stack. If the depth is $< 2$, the sequence is either continued by choosing one of the open brackets uniformly if the stack is empty or by choosing uniformly between open brackets and the closing bracket if the stack is non-empty. After $N - 2$ steps, the generated string could be a well-balanced string of length $N - 2$ in which case we add a depth 1 string of length 2 at the end which results in a well-balanced string of length $N$. Otherwise, the stack could be nonempty we add the remaining closing brackets which also leads to a string of length $N$.

To generate negative examples, we first sample a positive example and then corrupt some of the symbols in the string. For strings of length $N$, we first sample a number $k = 1, \ldots, N/10$ uniformly at random and then pick $k$ different indices uniformly at random. For each of those positions, with probability $1/2$, we swap the types of brackets, e.g. round '(' to square '[', and with probability $1/2$ we switch open brackets to closing brackets (or vice versa). There is a very small probability that after the corruption the resulting string will still be a valid well-balanced string of depth at most 2, in which case we redo the corruption again. The probability of the event is too low to affect the efficiency of the generation process.

**Additional Experiment with Mamba.** We explore how the size of the hidden state influences the performance of a Mamba model across various lengths on the Index Lookup task. We evaluate two-layer models of different widths $\{32, 64, 256, 512, 1024\}$ across various lengths ranging from 20 to 400. We find a clear trend where the performance of Mamba models increases monotonically with the increase in the width of the model (see Figure 3). While the performance does exactly scale linearly with width it is still somewhat interesting that the trend exists. We did a similar experiment with LSTM but did not observe such a trend and for lengths above 100 the performance remained at chance level even when the width was increased. Note, however, that even with a width of 1024, the performance of Mamba is still much worse than a one-layer Transformer with a width of 64.

## H.2 String Equality Task

We explore a few different strategies to evaluate the performance of models on the string equality task. In these experiments, the goal of the models is to determine whether the first half of the input string and the second half of the string are equal. The first two experiments are in the standard classification setting but differ in the way the negative examples are created. The third experiment is in the next character prediction which is commonly used in prior works to test models on formal languages. For the equality task, we find that it is not straightforward to generate negative examples in a way that the problem cannot be solved using shortcuts. Hence, we include the next character prediction setting which does not require the generation of negative examples.

**Summary of results.** The results on the String Equality task are relatively more nuanced than the results for index lookup and Dyck-2. On a standard binary classification setup, one must make a design choice regarding the generation of negative examples that could influence the difficulty of the task. We first conduct experiments in the classification setup by generating negative examples using two different strategies. On one task, we find that while recurrent models like LSTMs struggle beyond a certain length, state-space models like DSS and Mamba are able to match Transformer's performance. In the second task, we find that all models are able to achieve near-perfect accuracy. We note that both classification tasks can be solved by using shortcuts based on the way the negative examples are created. Hence, we explore another strategy called the next character prediction setting inspired by prior works [57, 20] on empirical analysis on formal languages. We note that the task in that setting becomes almost identical to the copying task where a model observes a sequence and then has to produce the same sequence. In that setting, we observe that at a small width like 64, state-space models fail to perform the task accurately at certain lengths whereas Transformers with the same width succeed. For models with larger widths, we find that DSS and Transformers succeed at performing the tasks for the lengths considered in our experiments.

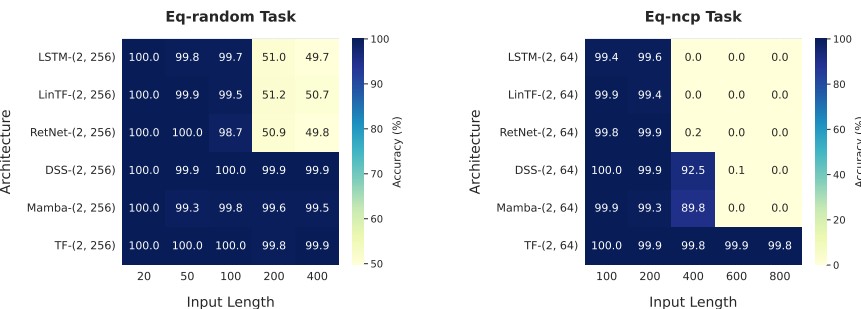

Figure 4: Performance of architectures on the Equality task. See Section H.2 for more details.

**Setup.** In all our experiments with string equality, we have a vocabulary $\Sigma$ which contains one token that is reserved as a separator token '[SEP]' and is placed between the first half and the second half of the input string. For each task, while creating an example we first pick the length $k$ uniformly at random from $\frac{N}{10}, \frac{N}{10} + 2, \frac{N}{10} + 4, \ldots N$. The choice of lengths is due to the requirement that the length of the strings must be even. During evaluation, we test the models on strings of length exactly $N$.

**(i) Eq-random.** The first experiment is pretty straightforward and contains binary strings, i.e., $\Sigma = \{0, 1, [\text{SEP}]\}$. With probability $1/2$, we create a positive example by first sampling a string of length $k/2$ uniformly followed by the separator token and then the same string again. By construction, the created example has a positive label. With probability $1/2$, we create the second half of the string by sampling another string of the same length where each symbol is sampled uniformly at random. We assign the label based on whether or not it is equal to the first half but with high probability $(1 - \frac{1}{2^{k/2}})$, the example has a negative label.

**(ii) Eq-one** In the second experiment, we have a vocabulary with $1024$ and we create the positive example in the same way as earlier in the Eq-one setting by sampling strings uniformly at random. To generate a negative example, we set the second half to be equal to the first half and then change the symbol at only one of the positions. In other words, the second half matches the first half at all positions except one.

The training details and hyperparameters are almost identical to the experiments described earlier.

**Results.** On both the Eq-random task and Eq-one task we find that Transformers achieve near-perfect accuracy on lengths up to 400. Recurrent models like LSTMs and linear Transformers struggle at lengths beyond 100 on the Eq-random task. On the other hand, we find that recently proposed state-space models like DSS and Mamba are able to match Transformers' performance on both tasks (See Figure 4 left). On the Eq-one task, we find that for lengths up to 400, all models are able to achieve near-perfect accuracy.

**Remark.** One thing to note is that both of the experimental setups described above can be solved using some form of shortcuts. In the first case, it is straightforward to see that the first half and the second half will differ at about half the positions on average. A recurrent algorithm does not necessarily have to store the first $N/2$ elements to compute the output correctly with high probability. Even if it stores the first few elements, then with high probability it can determine the label of the string correctly. For the second case (Eq-one), even if it might seem difficult at first look, a recurrent model can solve it perfectly by just maintaining a dictionary of the counts of each symbol in the vocabulary. Since the first half and second half differ at exactly one position, the number of occurrences of at least one symbol will be different in the two halves.

To rule out such phenomena, we adopt the next character prediction setting [20, 50, 57, 17].

**(iii) Eq-ncp.** In the next character setting (NCP), a model is required to predict the next set of valid continuations for every prefix of a given string. It can be seen as a multi-label classification problem for every prefix of a given string. The prediction for a particular input string is considered correct if the prediction for every prefix is correct.

In the context of the string equality task with length $N$, for the first $N/2$ symbols, all symbols are valid continuations, and hence the predictions for the first half of the string are trivial. After observing

the separator token [SEP], only one of the $|\Sigma|$ symbols is allowed at every prefix until the end of the string. We note that the prediction problem becomes equivalent to copying [28] a sequence of symbols. For our experiments the vocabulary size $|\Sigma| = 1024$. We explore sequences of higher lengths up to 800. In this setting, we find that when the model sizes are restricted, i.e., the width of the models is 64, state-space models such as DSS and Mamba struggle to perform better than chance-level accuracy for lengths over 400. In contrast, Transformers are able to achieve near-perfect accuracy (See Figure 4). However, unlike the case of Index Lookup, we find that the DSS architecture in particular is able to solve the task for lengths up to 800 with larger widths in the NCP setting.

**Discussion.** We discuss a few takeaways from our experiments. For the Index Lookup task, our results indicate that even small-sized one-layer Transformers can learn a lot more efficiently than recurrent models of much larger sizes. The experiments with bounded Dycks are primarily for one-layer Transformers and indicate that they learn at a much slower rate than recurrent models like LSTMs and even two-layer Transformers. On the string equality task, the difference in performance between Transformers and recurrent models is not as stark as the Index Lookup task, particularly with state-space models such as DSS. However, unlike Transformers, they seem to struggle on long sequences in the NCP setting when the widths of models are small.

