# OpenReview forum: "Separations in the Representational Capabilities of Transformers and Recurrent Architectures"
_NeurIPS.cc/2024/Conference — NeurIPS 2024 poster_

### Official Review · Reviewer_BJPt · 2024-07-09

**Soundness:** 3
**Presentation:** 4
**Contribution:** 3
**Rating:** 7
**Confidence:** 3

**Summary:**

This paper demonstrates theoretical separations in the representational abilities of Transformer and Recurrent Architectures on selected synthetic tasks, including index lookup, nearest neighbor, recognizing bounded Dyck languages, and string equality. The class of recurrent architectures examined includes popular networks such as RNNs, State Space Models, and Mamba. The results reveal significant differences in the required model sizes for each architecture to effectively represent these tasks. Additionally, the authors present some experiments that highlight the optimization efficiencies of architectures belonging to both classes on the selected tasks on practical-size sequences.

**Strengths:**

1) This is a theoretically solid paper. The proof techniques are novel and well explained, and they might be useful for further analysis of modern deep learning architectures.

2) The architectures considered are of practical interest, and the analysis contributes to a deeper understanding of the power and limitations of these architectures.

3) The paper extends previously known separation results to a set of diverse and more real-world applicable tasks.

**Weaknesses:**

1) The experimental part is fairly brief. The analysis of computational or statistical learning complexity is left to future work.

**Questions:**

- Do you have any guess of which Boolean functions are conjectured to be hard to compute for 2-layers Tranformers of poly-logarithmic size?

- It seems your separation results are worst-case over the input space. Can you argue whether all/some separations still hold for typical inputs, if for instance a uniform distribution over the input space is assumed?

**Limitations:**

Limitations and societal impact are well addressed.

---

> ### Author Rebuttal · Authors · 2024-08-06
>
> Thank you for your thoughtful comments and time.
>
> Responses to the individual comments below.
>
>
>
> > “The experimental part is fairly brief. The analysis of computational or statistical learning complexity is left to future work.”
>
> We respectfully disagree with this assessment. The central claims of the paper are theoretical in nature and we believe the empirical results are sufficiently extensive to support them and provide some context for the theoretical results.
>
> Our work does include experiments for all primary tasks considered in the paper – apart from those related to associative recall, where extensive empirical evidence has already been provided in earlier works (see citations in the paper). Additionally, we consider five different recurrent architectures spanning traditional architectures such as LSTMs to more recently proposed ones such as Mamba, DSS, etc. Given the kind of statements made in the theoretical results, it seems natural to explore with experiments whether differences of a similar nature are exhibited in practice. We believe we have studied that fairly extensively. While it is true that experiments analyzing the statistical learning complexity of models are not present, we believe that it is far and somewhat detached from the central claims of the paper.
>
>
> > “Do you have any guess of which Boolean functions are conjectured to be hard to compute for 2-layers Tranformers of poly-logarithmic size?”
>
> We do not have any strong conjectures at the moment, but there are a couple of directions worth thinking about.
>
> - In our paper, we show that 2-layer Transformers with poly-log size can represent k-DNFs/k-CNFs with at most N terms. We suspect that for k-DNFs/k-CNFs with cubic ($\Omega(N^3)$) or more terms may be difficult for two-layer Transformers to represent with poly-log size.
> - Due to circuit complexity bounds, functions outside of the complexity class TC0 are impossible for transformers even of polynomial size, and thus in particular for 2-layer transformers of poly-logarithmic size. This is likely to include foundational problems such as the evaluation of Boolean formulas. However, it is unclear if such tasks can represented by small-sized RNNs/SSMs.
>
>
> > “It seems your separation results are worst-case over the input space. Can you argue whether all/some separations still hold for typical inputs, if for instance a uniform distribution over the input space is assumed?”
>
> Thanks for raising this question. We have some thoughts on this. Currently, most of the results in our paper are of a deterministic nature in the sense that a model is said to express a function $f$ if it can produce the same output as $f$ for all inputs in the domain of $f$. This is more typical when analyzing expressivity in the context of classification, i.e., $f: X \rightarrow \{0, 1\}$.
>
> Upon some thinking over the past week,  we believe that some of our lower bounds can indeed be extended to the average case on certain distributions, such as the uniform one. For instance, there is a well-studied Boolean function called Inner-product mod 2 (IP2 in short), for which we can show that any RNN or 1-layer Transformer must have linear size to achieve low error with respect to the uniform distribution. The proof follows from the arguments in Section C.2 in Vardi et. al [1]. We think that some of the lower bounds in our paper, such as Index Lookup and nearest neighbor/MQAR, can be extended to various distributions based on reductions from the IP2 problem. We will verify the proofs and include them in the next version of our paper.
>
>
> [1] Vardi et. al. Size and Depth Separation in Approximating Benign Functions with Neural Networks. COLT 2021.

---

> > ### Comment · Reviewer_BJPt · 2024-08-09
> > **Thank you**
> >
> > I appreciate the authors' thoughtful response, and I will be raising my score accordingly.

---

### Official Review · Reviewer_hDB9 · 2024-07-11

**Soundness:** 4
**Presentation:** 3
**Contribution:** 2
**Rating:** 7
**Confidence:** 4

**Summary:**

This works studies the differences between Transformers and recurrent models with respect to 4 tasks: index lookup, associative recall, string equality and Bounded Dyck languages. The authors prove that for index lookup and nearest neighbor recall, there exists a 1-layer Transformer that needs poly-logarithmic width in the number of inputs while recurrent architectures require at least linear width. On the other hand, 1-layer Transformers require at least linear width for checking string equality and Bounded-Dyck languages. The authors strengthen their claims with experiments on synthetic datasets.

**Strengths:**

I found the following techniques were interesting: (1) the usage of Johnson-Lindenstrauss lemma to construct solutions with poly-logarithmic width in the number of inputs. (2) Using results from communication complexity to prove results for the lower-bound for the model width. While the ideas relating relating to communication complexity have been used in Sanford et al. (which is acknowledged by the authors), I nevertheless found the presentation of the lower-bound results to be interesting. I also found the writing to be refreshing; the authors are careful when interpreting the results and do not overstate the importance of their results while being precise in their claims.

**Weaknesses:**

Overall, I worry that the contributions of the work don't make progress towards the motivating questions in the introduction: why Transformers have supplanted  LSTMs (L29) in many problems. The analysis is specific to synthetic tasks with 1-layer recurrent models and Transformers (https://arxiv.org/abs/2105.11115, https://arxiv.org/abs/2208.01066). The theorems comment on the width of 1-layer Transformers but they do not seem to have any relevance to Transformers with multiple layers or when trained on other tasks.

With the goal of differentiating Transformers from LSTMs, I wonder why these particular tasks were chosen and what makes them important. It would help to motivate why these tasks are important and for example, explains why Transformers are better for language modelling. The different tasks are often motivated for specific contexts such as in-context learning (associative recall) or natural language itself (Dyck languages - https://arxiv.org/abs/2105.11115). Furthermore, the results also suggest that scaling up both Transformers and recurrent networks suffices for tackling any of the tasks.

Another concern is that the results are primarily for 1/2-layer Transformers. However, it is more relevant to study Transformers with multiple layers and it is unclear how to extend these results or if they have any implications for deeper Transformers. For example, we know that Dyck-languages can be represented by deeper Transformers.

**Questions:**

To reiterate the points raised above
1. How do we extend these results to deeper networks?
2. Are the results relevant for other task or other domains and is the relevance of these results very narrow?
3. Is there a way to motivate the selection of the tasks considered in this work?

**Limitations:**

The authors sufficiently acknowledge limitations. In particular they discuss that existence proofs do not guarantee such a solution is always found using gradient descent. The authors also comment on the limitations of their results to 1-layer Transformers and how they do not extend to 2-layer Transformers.

---

> ### Author Rebuttal · Authors · 2024-08-06
>
> Thank you for your thoughtful comments and time.
>
> > “The analysis is specific to synthetic tasks with 1-layer recurrent models and Transformers …”
>
> > “How do we extend these results to deeper networks?”
>
> We wish to clarify what may be a misunderstanding here – we should have been more explicit about this when we defined recurrent models on pg. 3. While our lower bounds for Transformers apply only to 1-layer models, our results for recurrent models apply to any depth. The way we have defined RNNs (Section 2), the transition function is allowed to be an arbitrary function which is very general and includes multi-layer versions of all architectures used in practice. When we state that an RNN needs a hidden state of size $\Omega(N)$ to solve a task, what we mean is that for any RNN with L layers, the sum of the size of hidden states of all layers has to be $\Omega(N)$ and thus the size of the model is $\Omega(N)$. Hence, our results show some tasks that can be solved with 1 or 2-layer Transformers (e.g. Index Lookup, Equality, Nearest Neighbour, etc) with poly-log size whereas any **multi-layer RNN** must have at least **linear size** (exponentially larger) to solve them. We will emphasize this point more clearly in Section 2 to avoid this confusion.
>
> We agree with the reviewer that understanding the limitations of multi-layer Transformers (TF) is very important. However, proving lower bounds even for 1-layer TF has been challenging—there have been some very recent results [4] and most questions about (multi-layer) TF with sub-linear width remain open. Our lower bound results for 1-layer TF are more general and more interestingly apply to tasks solvable by small-sized RNNs or small-sized 2-layer TF, thus showing a clear separation. Existing lower bounds for multi-layer TF are not solely information-theoretic and rely on unproven conjectures, e.g. from circuit complexity.  The situation mirrors somewhat that in computational complexity, where proving circuit lower bounds even for depth 2 or 3 has proved challenging for decades (https://shorturl.at/6lwUh).
>
> Even for 1-layer Transformers (TF) we believe there are many interesting unanswered questions, some of which are answered in our work. (Q1) Are there tasks that cannot be solved by small 1-layer TFs but are solvable by much smaller 2-layer TF?  (Q2) Since 1-layer TFs can solve tasks like Index lookup with poly-log size, while any multi-layer RNN needs linear size, are there tasks that small RNNs can solve but not 1-layer TFs?
>
>  (1) We show an exponential depth separation for tasks like Equality, Dyck, and Disjointness, where 1-layer TFs need linear width but 2-layer TFs solve them with log width. Such depth separation questions are widely studied for feedforward networks (see [1, 2, 3]). As far as we know, our results are the first to show exponential depth separation between 1-layer and 2-layer TFs on multiple tasks.
>
> (2) Since 1-layer TFs of poly-log size can express tasks that multi-layer RNNs with sublinear size cannot, the lower bound on Dyck shows a task solvable by constant-sized RNNs but not by 1-layer TFs.
>
> Regarding upper bounds or constructions for 1 or 2-layer TF, we see it as a strength rather than a limitation. We show multiple tasks that 1 or 2-layer TF can represent, whereas any multi-layer RNNs must be much larger regardless of depth.
>
> [1] The Power of Depth for Feedforward Neural Networks.
> [2] Depth Separation for Neural Networks.
> [3] Representation benefits of deep feedforward networks.
> [4] Representational strengths and limitations of transformers.
>
> > “With the goal of differentiating .. why these particular tasks were chosen and what makes them important …  for language modelling.”
>
> > “Is there a way to motivate the selection of the tasks in this work?”
>
> We discussed the task choices in the Introduction and further for Dycks in Section F. Briefly, our goal was to focus on tasks that appear natural or have strong justification in ML or linguistics research. For instance, in the MQAR task introduced in [1], they observed a perplexity gap in natural language modeling between Transformers (TF) and non-TF LMs, finding that non-TF language models struggled with texts that precisely mimic the MQAR task. Similarly, for the general version, i.e., the nearest neighbor task, multiple works [2, 3] found TF-based LLMs can mimic such behavior in practice.  Dyck languages have been extensively studied in formal language theory and ML research. Tasks like String Equality and Index Lookup are natural tasks that are likely to be relevant as primitives in search and learning tasks. For example, in code execution, given `arr = [6, 2, 8, 2, …, 3]` followed by `print(arr[i])`, the task is essentially Index Lookup. Models might also perform such lookups when provided with line numbers for debugging.
>
>
> [1]  arxiv.org/abs/2312.04927
> [2] arxiv.org/abs/2310.03016
> [3] arxiv.org/abs/2404.11018
>
>
> > “The analysis is specific to synthetic tasks…”
>
> > “Are the results relevant for other task or other domains and is the relevance of these results very narrow?”
>
> In the main paper, we discuss upper bounds for Transformers for specific tasks relevant to practice. In Section F, we show Transformers can represent a broader class of Boolean functions which include subclasses like k-CNFs/k-DNFs. Our lower bounds for multi-layer RNNs and 1-layer Transformers apply to any function with communication complexity ~ N, including most Boolean functions. Since the work aims to understand the problems theoretically, the problems must be mathematically well-defined and hence synthetic by definition. We're unsure what is meant by "other domains," but as these models are applied to more domains beyond NLP, like formal mathematics or algorithm design, understanding their capabilities/limitations along the lines we have considered in our paper would be imperative.  We hope the techniques/framework presented in our work will be useful for such analyses.

---

> > ### Comment · Reviewer_hDB9 · 2024-08-08
> > **Thank you for the rebuttal**
> >
> > Thank you for the detailed response, particularly on the challenges of extending these results to multi-layer Transformers and for clarifying that the setup includes multi-layer RNNs. Most of my concerns were addressed and I am hence raising my score.

---

> > ### Comment · Reviewer_LT9S · 2024-08-12
> > **Potential useful reference**
> >
> > I only followed the above discussion cursorily and there is a recent work that tries to provde lower bounds on depth for the transformer as a function of the Markovian order of the input process. Sharing it here in case it's of interest/relevance to the authors/reviewers: https://arxiv.org/abs/2407.17686v1

---

> > > ### Author Response · Authors · 2024-08-13
> > >
> > > Thanks for the pointer! Upon a quick look, it seems they also show communication complexity-based lower bounds for 1-layer Transformers and their lower bounds for multi-layer TF are based on some assumptions on the attention patterns. We will take a closer look and consider citing it appropriately in the next version.

---

### Official Review · Reviewer_FtyK · 2024-07-13

**Soundness:** 4
**Presentation:** 3
**Contribution:** 4
**Rating:** 8
**Confidence:** 4

**Summary:**

This paper analyzes the differences in terms of representations between Transformers and recurrent architectures. They highlight multiple cases: a) a setting where 1-layer Transformer can represent the task with a log number of parameters but not RNNs (index lookups) b) a case where RNNs can represent the task with a log number of parameters but not a 1-layer Transformer (Dyck language) c) a case where both 1-layer Transformer and RNNs cannot represent the task (boolean tasks like string equality). Then finally show that a 2-layer Transformer can represent the string equality task and other associative recall tasks such as nearest neighbors. They end up with some experiments to validate their theorems: they show that 1-layer transformers are great at learning the index lookup task and that RNNs learn very quickly the bounded dyck languages.

**Strengths:**

I think that the paper is of a great value to the community. There have been many papers in the community that the study the cases where  Transformers are superior over RNNs [1] and where RNNs are superior to Transformers [2]. It is great that this paper gives a unified picture of the strengths and weaknesses of both models. Besides, their analysis is clean: the theorems and proof sketch are very easy to follow. For this reason, I advocate for acceptance for this paper.




[1] Jelassi, S., Brandfonbrener, D., Kakade, S. M., & Malach, E. (2024). Repeat after me: Transformers are better than state space models at copying. arXiv preprint arXiv:2402.01032.
[2] Liu, B., Ash, J., Goel, S., Krishnamurthy, A., & Zhang, C. (2024). Exposing attention glitches with flip-flop language modeling. Advances in Neural Information Processing Systems, 36.

**Weaknesses:**

There are not a lot of flaws for this paper in my opinion. I will just give some suggestions:

- I believe that the authors could further improve the presentation of their results. In particular, it is not clear at the beginning why they choose the tasks they propose and it sounds a bit like a "catalog". I think the authors should clearly say that they consider a case where a 1-layer transformer with log(L) params fail but not RNN, another case where RNN with log(L) params fail but not a 1-layer etc. Maybe it may be worth adding a table where the column headers are the tasks and the row headers are the two models and each entry contains the complexity of each model at each task.

- Besides, I liked a lot the discussion from lines 210 to 218 and I believe that this should be earlier in the paper (even in the introduction). I think this point is central to understand one difference between Transformers and RNNs. I think that the point raised by the authors is not totally novel since a similar behavior has been reported by [1] in the case of the copy task.

- Regarding the superiority of RNNs over Transformers in the case of the bounded Dyck languages, do the authors have a similar discussion to add? If yes, could they add it? Is the explanation similar to the one advanced by [2]?

- Lastly, I don't know if it is possible to give any intuition about the limitation of 1-layer Transformers and RNNs at representing boolean functions?

- I am not sure to follow the experiments for the bounded Dyck languages. The authors say that the 1-layer Transformer achieve near perfect accuracy up to lengths 100 but the red curve in figure 2 right is around 60% for all the iterations. Did I miss something?


[1] Jelassi, S., Brandfonbrener, D., Kakade, S. M., & Malach, E. (2024). Repeat after me: Transformers are better than state space models at copying. arXiv preprint arXiv:2402.01032.
[2] Liu, B., Ash, J., Goel, S., Krishnamurthy, A., & Zhang, C. (2024). Exposing attention glitches with flip-flop language modeling. Advances in Neural Information Processing Systems, 36.

**Questions:**

I mentioned my questions in the weaknesses section.

**Limitations:**

The authors mention the limitations of their work.

---

> ### Author Rebuttal · Authors · 2024-08-06
>
> Thank you for your thoughtful comments and time.
>
> Responses to the individual comments below.
>
> > “I believe that the authors could further improve the presentation of their results. In particular, it is not clear at the beginning why they choose the tasks they propose and it sounds a bit like a "catalog". I think the authors should clearly say that they consider a case where a 1-layer transformer with log(L) params fail but not RNN, another case where RNN with log(L) params fail but not a 1-layer etc. Maybe it may be worth adding a table where the column headers are the tasks and the row headers are the two models and each entry contains the complexity of each model at each task.”
>
> Thank you for the suggestion. We agree that categorizing with respect to the separations (log L vs L) could be more helpful. We considered adding a table/figure at the top of the second page. We weren’t sure about the best way to depict the results and due to lack of space, we decided to go without it. We will consider adding these in the next version.
>
>
> > “Besides, I liked a lot the discussion from lines 210 to 218 and I believe that this should be earlier in the paper (even in the introduction). I think this point is central to understand one difference between Transformers and RNNs. I think that the point raised by the authors is not totally novel since a similar behavior has been reported by [1] in the case of the copy task.”
>
> Thanks for the suggestion and we will consider adding it to the introduction in the next version. Regarding your point about [1], we do not claim novelty about the intuition. We believe other researchers in the community may have a similar intuition behind the differences in the two architectures. In our work, that intuition is formalized using communication complexity for lower bounds and the JL vectors for upper bounds. In their work [1], they use a different approach to derive a lower bound for the copying task. However, it is worth noting that using communication complexity-based techniques as used in our work, it is also possible to show lower bounds for RNNs to perform the copying task. We will consider adding that somewhere in the appendix for readers who might find it interesting.
>
> > “Regarding the superiority of RNNs over Transformers in the case of the bounded Dyck languages, do the authors have a similar discussion to add? If yes, could they add it? Is the explanation similar to the one advanced by [2]?”
>
> We have some discussion on this at the beginning of Section F, which you might find helpful. However, the discussion is intertwined with the proof technique, so it might be less straightforward than the former discussion. The general intuition is that bounded Dycks can be represented by constant-sized DFAs, implying that RNNs require very little (constant) memory while processing them sequentially. In contrast, a 1-layer Transformer must compute the same based on a convex combination of all input symbols (attention) and classify 0 or 1 based on this vector (using MLP). The weights of the convex combination are determined by dot products of query and key projections, and hence cannot be arbitrary.
>
> In simpler terms, the proof intuitively shows that if a 1-layer Transformer can recognize bounded Dyck languages, then the query and value vectors must contain sufficient information, $\Omega(N)$ in bits. Think of it this way: the width of the query/key vectors determines the flexibility of the attention weights, and the width of the value vectors or input embeddings determines the amount of information stored in the output of the attention block. The proof essentially shows that both must be sufficiently wide (contain enough bits of information) for any arbitrary MLP block to be able to classify correctly based on the attention block's output.
>
> > “Lastly, I don't know if it is possible to give any intuition about the limitation of 1-layer Transformers and RNNs at representing boolean functions?”
>
> For RNNs, even for Boolean functions, the key intuition is the same as the one explained for the Index Lookup task. The goal of introducing that task was to highlight that intuition in a natural manner since it may be less obvious in the context of Boolean functions. For 1-layer Transformers, the intuition is the same as the one described above.
>
> > “I am not sure to follow the experiments for the bounded Dyck languages. The authors say that the 1-layer Transformer achieve near perfect accuracy up to lengths 100 but the red curve in figure 2 right is around 60% for all the iterations. Did I miss something?”
>
> For bounded Dyck languages, we find that 1-layer Transformers struggle at lengths 200 and above, unlike RNNs and 2-layer Transformers. Figure 2 right depicts the validation curve for models at length 400 (and not 100). The goal of that line about lengths 100 is to clarify that this difference is performance occurs at lengths above 100 or so which is natural since the difficulty of the tasks increases with length. For the Index Lookup task, all models solve the task at length 20 and the difference is more stark at length 100 and above (see Figure 2 left).

---

> > ### Comment · Reviewer_FtyK · 2024-08-09
> >
> > I thank the authors for their reply. They properly answered to all my questions.  I maintain my score.

---

### Official Review · Reviewer_LT9S · 2024-07-16

**Soundness:** 4
**Presentation:** 4
**Contribution:** 3
**Rating:** 7
**Confidence:** 4

**Summary:**

In this paper, the authors study the representational separation results about two widely used classes of language models: transformers and RNNs. For a set of practically well-motivated tasks, they establish lower and upper bounds for attention based one-layer (and some for two) transformers and arbitrary RNNs using and building upon techniques from communication complexity. They also empirically validate their conclusions.

**Strengths:**

First of all, I want to appreciate the authors for clearly elucidating the main ideas and the intuitions in a very accessible manner even for non-experts. Such style of writing is a rarity these days and the authors deserve the credit for this.

Overall, it's a very important topic and problem of research interest to understand the differences between the two widely used models in the form of transformers and RNNs. They present a set of intereseting results for well-motivated tasks, and establish theoretical results for transformers needing to be of linear size in the input dimension, whereas RNNs could get away with logarithmic size, and vice versa.

**Weaknesses:**

While I appreciate the technical results of the paper, I am not fully sure about how meaningful these results are inorder to decipher the fundamental differences between RNNs and transformers. In particular, for the index lookup task, it makes sense that the only way RNNs can retrieve the symbol $s_p$ at an unknown location $p$ revealed at the end is only via storing all the past information in its hidden state and hence $m \geq N/p$ is logical. However, if you feed the sequence in the order $p, s_1, \ldots, s_N$, would the same result hold? I believe you could get away with $\log N$ here too though I could be wrong. Shedding light on things like this could yield more insights about how these architectures are fundamentally different? Because for an index lookup task, if all we care about is retrieving a symbol at some position, we don't really care about which order you feed the input sequence right?

Also on a minor note, as the authors themselves acknowledged, these results might not have full bearing on the learnability settings. It would be interesting to see how these results hold for SGD learnability on the same tasks. Food for future thought.

**Questions:**

1. I realized that the soft-max attention in line 121 is non-causal. Would your results change if it's causal attention?

**Limitations:**

Yes.

---

> ### Author Rebuttal · Authors · 2024-08-06
>
> Thank you for your thoughtful comments and time.
>
> Responses to the two points raised in your review below.
>
> > “In particular, for the index lookup task, it makes sense that the only way RNNs can retrieve the symbol $s_p$ at an unknown location revealed at the end is only via storing all the past information in its hidden state and hence $m\geq N/p$ is logical. However, if you feed the sequence in the order $p, s_1,\ldots, s_N$, would the same result hold? I believe you could get away with $\log N$ here too though I could be wrong. Shedding light on things like this could yield more insights about how these architectures are fundamentally different? Because for an index lookup task, if all we care about is retrieving a symbol at some position, we don't really care about which order you feed the input sequence right?”
>
> While what you have mentioned is technically correct, that approach is specific to the particular version of the Index Lookup task and it does not overcome the fundamental issues with how RNNs process inputs.
>
> To be clear, for the case where the models are provided sequences in the order $p, s_1,\ldots, s_N$, RNNs can indeed get away with a hidden state of size $\log N$. However, consider the following variant of the Index Lookup task (let’s call it Multi-Index Lookup) where the models are provided with $k =O(N)$ (e.g. N/2, N/4, etc)  indices for which they have to look up and produce the respective symbols. In particular, the input is $s_1, \ldots, s_N, p_1, p_2, \ldots, p_{k}$.  For each $p_i$ after the symbols $s_1, \ldots, s_N$, the model is required to output the respective symbol at position $p_i$. Note that for one-layer Transformers, the construction for the Index Lookup task can be directly extended for this Multi-Index Lookup task implying that poly-log size Transformers can solve this.
>
> However, for RNNs, they must have a hidden state of size $\Omega(N)$ to solve this correctly even if the positions are prepended to the symbols. Instead of a reduction from the $\mathrm{Index}$ problem, we can show via a simple reduction from the Disjointness problem. In other words, if the input sequence is of the form $p_1, p_2, \ldots, p_{k}, s_1, \ldots, s_N, p_1, p_2, \ldots, p_{k}$ and an RNN can produce the required outputs then two parties Alice and Bob can compute Disjointness using $mp$ bits which implies that $mp$ must be $\Omega(N)$.
>
> The argument is quite straightforward. Let’s say Alice and Bob have two bits strings $a$ and $b$ of size $k=O(N)$ and both have access to the same RNN. For simplicity, they have agreed that Alice will place her bits in the first $k$ indices sequentially. Alice can then take an RNN, and provide it with indices $1$ to $k$ as input followed by $k$ symbols corresponding to $a$. The last $N - k$ symbols can be arbitrary. Alice can then send the hidden state to Bob which requires $mp$ bits. Bob can provide the same set of indices $1$ to $k$ to the RNN and then record the output. Bob can then compute Disjointness and hence it follows that $mp= \Omega(N)$.
>
> **Summary:** While it is true that for the Index Lookup task prepending the input sequence with the position/indices can make a difference for RNNs, the limitation/lower bound still applies for a natural extension of the task involving multiple indices. For us, the goal of introducing the Index Lookup was to come up with the simplest task to describe our tools. However, based on the point raised by you,  we see that it could create confusion, and will make sure to clarify this in the next version of the paper.
>
>
>
> > “I realized that the soft-max attention in line 121 is non-causal. Would your results change if it's causal attention?”
>
> Using causal attention does not affect any of the constructions or lower bounds for Transformers in the main paper, i.e., Index Lookup, Equality, Disjointness, Nearest Neighbor, etc. It only affects the construction of Transformers presented in Section F.3 for a more abstract yet general class of Boolean functions.

---

> > ### Comment · Reviewer_LT9S · 2024-08-12
> > **Acknowledgement of the rebuttal**
> >
> > I am satisifed with the authors' response which addressed my concern. Happy to raise my score to 7.

---

### Author Rebuttal · Authors · 2024-08-06

We thank all the reviewers for their thoughtful feedback and their time. We are encouraged to see that they found our results interesting (Rev *LT9S, FtyK, BJPt*), well-motivated (Rev *LT9S,  BJPt*), and to be of value to the community (Rev *FtyK*). We are further pleased to see that they found our proof techniques to be interesting (Rev *LT9S, BJPt, hDB9*), clean (Rev *FtyK*), and well-explained (*all reviewers*).

In this work, we show that various tasks can be solved by small-sized (poly-log size with 1 to 2 layers) Transformers, whereas any multi-layer RNN must be exponentially larger in comparison. Additionally, we show that 1-layer Transformers cannot solve certain tasks with sublinear width, whereas they can be solved by either RNNs or 2-layer Transformers of much smaller size.


We have addressed the weaknesses and specific questions from each reviewer in the individual responses. Below, we summarize the key aspects of our responses. Please refer to the individual responses for more details.


------------------------------------------------


Reviewer LT9S mentioned that the Index Lookup task for which we provide a linear lower bound for RNNs can be substantially easier if the inputs are provided in a different way. In the rebuttal, we explain why it does not solve the core issue and show that a natural extension of the task will still be hard for RNNs even if the inputs are provided in a different manner.

Reviewer FtyK provided some suggestions and had some questions, which we have answered in the individual response.

Reviewer hDB9 mentioned that our analysis is specific to 1-layer recurrent models and Transformers. We believe there may be a misunderstanding. We clarify that our lower bounds for recurrent networks apply to models of any depth, not just 1-layer RNNs. Further, we discuss the implications and relevance of the lower bounds for 1-layer Transformers and other questions raised by the reviewer in the individual response.

Reviewer BJPt stated that the experimental part of our paper is brief and does not include experiments to analyze statistical learning complexity. We argue that the experiments are supportive in nature and are sufficiently extensive to provide context for the theoretical results in the paper. We address the specific questions raised by the reviewer in the individual response.

---

### Decision · Program_Chairs · 2024-09-25

**Decision:**

Accept (poster)

**Comment:**

The work gives several theoretical separation results between Transformers and recurrent models. For example, the authors show that Transformers can implement index lookup, string equality and nearest neighbors, but RNNs require linear size to solve these problems. On the other hand, they show a lower bound for single layer Transformer on recognizing Dyck languages, which can be solved with RNNs.

As all reviewers gave a positive evaluation of this work, and agree that it is a strong submission, I recommend that it is accepted.